# Synthesis of Iron, Zinc, and Manganese Nanofertilizers, Using Andean Blueberry Extract, and Their Effect in the Growth of Cabbage and Lupin Plants

**DOI:** 10.3390/nano12111921

**Published:** 2022-06-04

**Authors:** Erika Murgueitio-Herrera, César E. Falconí, Luis Cumbal, Josselyn Gómez, Karina Yanchatipán, Alejandro Tapia, Kevin Martínez, Izar Sinde-Gonzalez, Theofilos Toulkeridis

**Affiliations:** 1Centro de Nanociencia y Nanotecnología, Universidad de las Fuerzas Armadas ESPE, Av. Gral. Rumiñahui s/n, Sangolquí P.O. Box 171-5-231B, Ecuador; lhcumbal@espe.edu.ec (L.C.); kemartinez2@espe.edu.ec (K.M.); 2Departamento de Ciencias de la Tierra y de la Construcción, Universidad de las Fuerzas Armadas ESPE, Av. Gral. Rumiñahui s/n, Sangolquí P.O. Box 171-5-231B, Ecuador; jpgomez6@espe.edu.ec (J.G.); lkyanchatipan@espe.edu.ec (K.Y.); alejo_tapia95@hotmail.com (A.T.); iisinde@espe.edu.ec (I.S.-G.); 3Departamento de Ciencias de la Vida y de la Agricultura, Universidad de las Fuerzas Armadas ESPE, Carrera Ingeniería Agropecuaria IASA 1, Av. Gral. Rumiñahui s/n, Sangolquí P.O. Box 171-5-231B, Ecuador; cefalconi@espe.edu.ec

**Keywords:** nanofertilizers, nanoparticles, Andean lupin, cabbage, Andean blueberry extract

## Abstract

The predominant aim of the current study was to synthesize the nanofertilizer nanoparticles ZnO_MnO-NPs and FeO_ZnO-NPs using Andean blueberry extract and determine the effect of NPs in the growth promotion of cabbage (*Brassica oleracea var. capitata*) and Andean lupin (*Lupinus mutabilis* sweet) crops. The nanoparticles were analyzed by visible spectrophotometry, size distribution (DLS), scanning electron microscopy (SEM), and transmission electron microscopy (TEM). Solutions of nanoparticle concentrations were applied to cabbage, with solutions of 270 and 540 ppm of ZnO_MnO-NPs and 270 and 540 ppm of FeO_ZnO-NPs applied to Andean lupin. Zinc was used in both plants to take advantage of its beneficial properties for plant growth. Foliar NPs sprays were applied at the phenological stage of vegetative growth of the cabbage or Andean lupin plants grown under greenhouse conditions. The diameter of the NPs was 9.5 nm for ZnO, 7.8 nm for FeO, and 10.5 nm for MnO, which facilitate the adsorption of NPs by the stomata of plants. In Andean lupin, treatment with 270 ppm of iron and zinc indicated increases of 6% in height, 19% in root size, 3.5% in chlorophyll content index, and 300% in leaf area, while treatment with 540 ppm of iron and zinc yielded no apparent increases in any variable. In cabbage, the ZnO_MnO-NPs indicate, at a concentration of 270 ppm, increases of 10.3% in root size, 55.1% in dry biomass, 7.1% in chlorophyll content, and 25.6% in leaf area. Cabbage plants treated at a concentration of 540 ppm produced increases of 1.3% in root size and 1.8% in chlorophyll content, compared to the control, which was sprayed with distilled water. Therefore, the spray application of nanofertilizers at 270 ppm indicated an important improvement in both plants’ growth.

## 1. Introduction

Increases in the world population and demand for food supplies have motivated scientists and engineers to devise new methods to boost and intensify modern precision agriculture [1,2,3]. However, due to the limited availability of land and water resources, increased productivity has been achieved by using shifting agriculture, through the effective use of modern technology [4,5,6]. Such an approach in agriculture has had repercussions on the biological diversity of the area, extending consumption over a larger area and reducing the time available for its regeneration [7,8,9]. It has also reduced the length of the fallow period, decreased soil fertility and yield, and forced farmers to extend their crops to forest land, causing alterations in the ecosystem [10,11,12]. In addition, farmers have regularly used conventional fertilizers, which mainly contain the macronutrients: N, P, and K. These compounds are synthetic products that appear in many forms, such as granular or liquid. Although conventional fertilizers are still widely used, they usually show low efficiency rates. Thus, large amounts of fertilizers are needed for a given yield and quality.

As a result, within the last few decades, conventional fertilizers had a dominant role in various environmental challenges due to their high application rates, low efficiency, and application methods. Furthermore, they disturb the mineral balance and cause a reduced availability of micronutrients; additionally, they may impact the soil dynamics, often resulting in irreparable damage to its structure and mineral cycles [13,14]. For instance, soil that is too alkaline or acidic prevents the plant’s roots from accessing nutrients, which can adversely affect plant growth. Increasing soil pH, especially above 6.5, results in the decreased extractability and availability of native or added forms of Zn, Fe, and Mn in plant’s soil [15,16]. Symptoms caused by nutrient deficiencies are generally purplish-red coloring, necrosis, stunted growth, chlorosis, or interveinal chlorosis [17]. To reduce the negative impacts of agriculture, as well as guarantee the timely and constant supply of food, the fortification of food production, with the input of nanofertilizers, is proposed [18,19,20]. Fertilizers, at the nanoscale dimension, might regulate the release of nutrients, deliver the right quantity of nutrients required by the crops in a suitable proportion, and promote productivity, while ensuring environmental safety [21].

In this regard, several studies that focused on the fabrication and use of nanofertilizers have been carried out, based on nutrients for crops with a variety of elements, mainly micronutrients such as Mn, Cu, B, Zn, Fe, Ni, and Mo [22,23]. From laboratory-scale research, it has been reported that these nanoparticles are able to contribute to crop productivity by improving the seed germination rate and seedling growth of radish (*Raphanus sativas*), lettuce (*Lactuca sativa*), corn (*Zea mays*), and cucumber (*Cucumis sativus*) [24,25,26,27]. A positive effect on the growth of mung (*Vigna radiate*) and chickpea (*Cicer arietinum*) seedlings has also been observed with an optimal concentration of nano-ZnO [28,29,30]. It is also known that several commercial products contain nanoparticles of zinc, silica, iron, titanium dioxide, and gold, among others [31,32,33]. However, due to the environmental concerns of synthesizing nanostructures with conventional methods that use toxic chemicals, there is currently a great interest in improving biosynthesis procedures that are easy to develop, economical, and environmentally friendly [34,35,36]. These techniques include a wide range of natural precursors, such as plant extracts, bacteria, and enzymes [37]. Specifically, the use of plant extracts for nanoparticle synthesis is potentially advantageous over microorganisms due to the cost of the cultures, short production times, biohazards, and ability to increase production volumes [38,39].

Manganese is an important microelement for the proper functioning of a plant’s metabolism, due to its influence on the photosynthetic capacity, as well as nitrogen assimilation, as a deficiency can cause great economic losses in crops of commercial interest [40]. The most common way to counteract this imbalance is with bulk KMnO_4_ or Mn_2_O_3_ supplements [41]; however, MnO nanoparticles emerge as an innovative alternative to be used as nanofertilizers. A study by [42] demonstrates how MnO nanoparticles at concentrations lower than 500 ppm stimulate root growth in lettuce seedlings, in addition to generating larger-sized seedlings; moreover, a lower toxicity than that of copper or zinc nanoparticles is evidenced. MnO NPs have a positive influence on the amount of germinated sprouts, as was seen in the case of [41], where they used MnO nanoparticles to stimulate the sprouting of watermelon seeds; thus, there is an increase in productivity in crops of commercial interest.

Iron nanoparticles have the potential for development in the field of agriculture. Although their main use lies in the decontamination of the environment, great advances have been observed in fertilization capacity, the most common forms in which iron nanoparticles are presented are Fe_2_O_3_, Fe_3_O_4_, and Fe [40]. In a study on peanut plants, an increase in root size and the amount of biomass was observed for treatments with 20 nm nanoparticles of γ-Fe_2_O_3_ (Raju, Mehta, and Beedu, 2016); similar results were observed in spinach plants using α-Fe_2_O_3_ with a size of 50 nm [43], while, in soybean crops, the photosynthetic capacity was favored [44].

The effectiveness of ZnO NPs has been proven individually in several research. In the application of different plants, it is known that ZnO NPs reduce the presence of diseases via their antifungal activity against *Penicillium expansum*, *Botrytis cinerea*, *Aspergillus flavus*, *Aspergillus niger*, *Aspergillus fumigatus*, *Fusarium culmorum*, *Fusarium oxysporum* [45,46,47], and antimicrobials for crop protection [48]. Their antifungal effect is partly because they cause malformation of hyphae, leading to fungal death [49]. According to [50], they improve plant growth and fruit quality by increasing the sugar concentration [51,52,53]. The authors of pointed out that the application of metallic NPs, such as zinc, has been shown to have significant effects on seed germination and plant growth. Other reports indicate a phytotoxic effect on different cultivated plants. Iron plays a significant role in enhancing plant yield, biomass, and growth by providing essential nutrients [54].

Zinc oxide nanoparticles (ZnO NPs), iron oxide nanoparticles (FeO NPs), and magnesium oxide nanoparticles (MgO NPs) have been prepared and applied to plants, in order to increase their agronomical and physiological traits. Based on the aforementioned context, the current study has aimed to synthesize multicomponent NPs of iron, zinc, and manganese nanofertilizers.

Andean blueberry (*Vaccinium myrtillus*) is an example of a plant with high antioxidant potential, due to its high level of flavonoids. It originated from regions of South America, where it is appreciated due to its exotic flavor and “medicinal power” (mainly related to longevity) [55]. Polyphenols are also found to be high in bark and pulp, which offer some protective effects to the cell walls [56]. Blueberry flavonoids have shown antioxidant activity, which can be used as photoprotector for 90 days [57]. Flavonoids undergo oxidation; during this process, when the organic compounds are exposed to light and high temperatures, one or more electrons are transferred to another substance (Silva et al., 2020). Additionally, the blueberry fruit contains large amounts of polyphenols, which are biodegradable and soluble in water at room temperature; they can be used as a reducing agent during the formation of nanoparticles and for stabilization [58,59].

Furthermore, some researchers have used industrial by-products, such as ashes, shells, and organic animal waste (i.e., manure), or adopted agroecology and organic production [60,61] to improve agriculture yields. Particularly, the sugarcane (*Saccharum officinarum* L.) bagasse (SCB) that contains approximately 75% of SiO_2_ [62] is the by-product of the sugar and ethanol industry. On average, 140 to 280 kg of bagasse are generated per ton of sugarcane [63,64,65]; thus, about 300 million tons of sugarcane bagasse is produced. This residual has been widely used for a number of industrial applications, such as fuel, cement-replacing material, production of glass-ceramic material, and geopolymers, among many others [66]. Therefore, the monetary value that could be recovered from sugarcane bagasse reuse might be attractive for investors. The final product of burning the sugarcane bagasse is the residual ash (SCBA), which is normally used as a fertilizer in sugarcane plantations because it contains SiO_2_, Fe, Mn, and Zn. Thus, by-products can be used as precursors of valuable materials for the industry. Researchers have also prepared a nanofertilizer that was extracted from banana peels under alkaline conditions. They found that the banana peel extract mainly contained elemental potassium chelated with citric acid. Additionally, they found other minerals, such as iron, magnesium, copper, sodium, calcium, and manganese, chelated with citric acid in the nanostructure. In addition, the nanofertilizer had a spherical morphology, with a major particle size of 40 nm, and showed great germination efficiency in the first planting week for both tomatoes and fenugreek [67].

On the other hand, the Andean lupin (*Lupinus mutabilis*) is a legume that has been domesticated and cultivated for more than 4000 years by the pre-Hispanic cultures of the Andean zone. Due to its good taste and protein content, the lupin seed contributes significantly to the food and nutritional security of the Andean population [68]. I-450 Andino is an Andean lupin cultivar that is bred based on earliness and agronomic traits [69]. Additionally, the cabbage (*Brassica oleracea* var. *capitata*) provides humans with food of low caloric density, but it is rich in fiber, vitamins, minerals, bioactive components, and secondary metabolites that favor health; additionally, it is suggest that facilitates the prevention of various types of cancer, cardiovascular diseases, and metabolic disorders [70]. The effect of nanofertilizers were studied in the growth of both plants.

The current study focused on the green synthesis of zinc (ZnO NPs), iron (FeO NPs), and magnesium (MgO NPs) oxide nanoparticles, using Andean blueberry extract as a reducing agent and attempting to reuse sugarcane bagasse ash, which is a residue from the sugar industry in Ecuador, to optimize the nutritional properties of the nanoparticles. In addition, the prepared nanoparticles were applied to cabbage and lupin in the form of a foliar spray, in order to benefit their growth under greenhouse conditions. To our knowledge, ZnO–MnO and FeO–ZnO nanoparticles prepared with Andean blueberry extract have never been applied as a foliar additive with sugarcane ash on cabbage and lupin plants, which supports the novelty of the present work.

## 2. Materials and Methods

### 2.1. Study Area

This research was conducted in a greenhouse and the field. The first area (Zone 1) was located at the IASA I Campus, within the Hacienda el Prado of the Universidad de las Fuerzas Armadas-ESPE, close to the city of Sangolquí in the inter-Andean valley of north-central Ecuador, which is at an altitude of 2748 m above sea level. There is an average annual temperature of 14 °C and humid mesothermal equatorial climate. Two trials have been conducted, i.e., in the greenhouse and field, with areas of 60 m^2^ and 240 m^2^, respectively [71]. The second study area (Zone 2) was located in the facilities of the Agro-Hidropónica San Antonio Cia. Ltda, within the parish of Uyumbicho, 11 km west of the first study area. The study area had an extension of 274 m^2^ (Figure 1).

### 2.2. The Andean Blueberry Extract Preparation

For the synthesis of the nanoparticles (NPs), the extract of Andean blueberry, a native wild shrub from Ecuador that is also called agraz (*Vaccinium floribundum*), was used. The fruit was selected according to its level of maturity, and only those with a reddish-black color were used. Then, it was subjected to cleaning and disinfection. The fruit was liquefied with 95% ethanol in a 2:1 ratio at 40 °C. Subsequently, it was left to rest for 48 h; then, it was filtered through filter paper and 450 µm filter, discarding the solid phase. The extract was separated from the ethanol with a rotary evaporator, i.e., the Buchi 850. The extraction of the fruit was followed by filtering the final solution through a 250 µm filter. Antioxidant capacity analysis was determined by the Folin–Ciocalteu method [58]. Following this, the NPs were characterized with dynamic light scattering (DLS), ultraviolet–visible spectroscopy (UV–VIS), transmission electron microscopy (TEM), and scanning electron microscopy (SEM).

### 2.3. Preparation of the Sugarcane Bagasse Ash

To prepare the ash, sugarcane bagasse from Uyumbicho town, central Ecuador, was used. The sugarcane was subjected to deep cleaning and disinfection; it was passed to a 50 °C stove, followed by 100 °C, with two h at each temperature. Samples were dried, crushed, and calcined on a muffle (HYSC MF-05, Seoul, Korea) at 500 °C for one h and sieved on a No. 200 sieve.

### 2.4. Characterization of Sugar Cane Bagasse Ash

The ash was dried in a Memmert (model SN30, Schwabach, Germany) oven; then, 1.0 g of ash was weighed on a Cole Palmer (model 1000016, Vernon Hills, IL, USA) analytical balance. The ash was suspended in 50 mL of distilled water and stirred; this solution was used to measure the pH and electrical conductivity using a Hach multiparameter meter (model HQ30d, CO, United States) [72]. To determine the hydrogen potential, it was calibrated with buffer solutions of 4, 7, and 9 pH. Resolution: selectable between 0.001 to 0.1 pH; pH electrode calibration: 0.002; pH measurement: 0 to 14 pH. The electrical conductivity was calibrated with a standard solution of 1413 µS/cm. Conductivity measurement range: 0.01 μS/cm to 200 mS/cm; conductivity resolution: 0.01 μS/cm with 2 digits; conductivity accuracy: ±0.5% from (1 μS/cm–200 mS/cm). Conductivity and pH were determined following methods 4500-H + B and 2510B, respectively, of the Standard Methods (18th edition), using the HQ30d multiparameter meter (Hach). Bulk density was determined using the given protocol [73,74]. Grain size and moisture content analyses were performed following the methodology established by Manals et al. [73]. Metals (Fe, Mn, and Zn) were determined by flame atomic absorption using the Aanalysit 800 atomic absorption spectrometer (Pelkin Elmer, Shelton, CT, USA), following the methodology detailed in the Standard Methods (18th Edition), using method 3111-B [72]. Iron, manganese, and zinc were determined by AA via the AAnalyst 800 PE equipment, following method 3111 B (direct air-acetylene flame method): Fe (%Rec. 101%, RDS: 2.35), Mn (%Rec. 99%, RDS 1.02), and Zn (%Rec 104%, RDS 2.35). RSD is the relative deviation.

### 2.5. Synthesis of Nanoparticles (NPs)

A variety of NPs, such as those of iron, zinc, manganese, and multicomponent nanoparticles, were synthesized. For the iron nanoparticles of FeO NPs, a 0.1 mol/L solution of ferrous sulfate heptahydrate was used (FeSO_4_.7H_2_O) (lot# AD 8158; Daigger Sci-Ed Warehouse—CAS 7664-93-9, IL, USA). To this solution, the Andean blueberry extract was added in a 10:1 ratio (100 mL of extract: 1 g of ash) and sonicated; then, the pH was slowly adjusted between 8 and 10 in an orbital shaker (model SHO-2D, Gangwon-do, Korea), with a 0.01 mol/L NaOH solution (Fisher Scientific, Hampton, NH, USA; Lot# 147439), by means of continuous dripping [58]. In the case of zinc nanoparticles (ZnO NPs), solutions of 0.1 mol/L zinc acetate dihydrate were prepared (Zn(O_2_CCH_3_)_2_(H_2_O)_2_) (Fisher Scientific, Hampton, NH, USA; lot# 930569) and 0.1 mol/L sodium hydroxide (NaOH) (Fisher Scientific, Hampton, NH, USA; lot# 147439). This solution was preheated at 40 °C, and the solution was dried at 70 °C for 12 h, after which it was subsequently calcined in a muffle (HYSC MF-05, Seoul, Korea) at 450 °C for 2 h and 20 min. The ZnO NPs were obtained, and Andean blueberry extract with ash was added in a 10:1 ratio and filtered with a 250 µm pore membrane [75]. Manganese nanoparticles (MnO NPs) were prepared with solutions of 0.1 mol/L manganese (II) sulfate monohydrate (MnSO_4_.H_2_O) (Loba Chemie, Mumbai, India, Lote # GM05061401) and 0.1 mol/L sodium hydroxide (NaOH) (Fisher Scientific, Hampton, NH, USA, lot# 147439), which were preheated to 60 °C. The pH of the solution was adjusted to 12, and the final solution was left stirring for 1 h to complete the reaction [76]. Andean blueberry extract with ash was added to the solution in a 10:1 ratio, then filtered through a 250 µm membrane filter.

Finally, multicomponent nanoparticles were prepared, mixing solutions of ZnO with MnO NPs and FeO with ZnO NPs, obtaining two multicomponent solutions of ZnO_MnO-NPs and FeO_ZnO-NPs. From the final NPs solutions, precipitates were discarded, and subsequently characterized. These were the ones that were used to apply in the cabbage (*Brassica oleracea* var. *capitata*) and lupin (*Lupinus mutabilis* Sweet) crops [77,78,79] (Figure 2).

### 2.6. Materials and Equipment Used in the Characterization of NPs

For extraction of the Andean blueberry, a YAMATO (model RE801, Flawil Switzerland) steam rotator was used, while, for the synthesis of the nanoparticles, an orbital shaker and laboratory material were utilized. This equipment has annual maintenance of its electronic parts. The accuracy and precision of the equipment was not achieved because it is limited to solvent extraction, which was not performed for the measurements. The characterization of the NPs was carried out, in order to determine their mean sizes and morphologies. Additionally, data were provided regarding the absorbance peaks at a given wavelength. Hereby, the characterization of the size distribution of the different nanoparticle solutions was performed with a HORIBA LB-550 submicron particle analyzer, which was connected to a computer (with a standard SCSI-type interface) that used the HORIBA software LB-550. (Dinamic Light Scattering Particle-Size Analizer (P1000797001b) Version 3.57; 1996-2005, Kyoto, Japan). The measurement was based on the principle of dynamic light scattering, i.e., to measure the size of the particles within a range between 0.001 and 6 μm or 1 and 6000 nm. This equipment was maintained, checked for alignment, and calibration annually. The calibration curve was performed using certified NIST-traceable latex (polystyrene and polydisperse) standards of 20 ± 2 and 100 ± 2 nm. The results were accepted with a 5% tolerance. In addition, repeatability and reproducibility calculations were performed.

The subsequent characterization of the particle size was performed by TEM. The protocol consisted of allowing a drop of the colloidal suspension of nanoparticles to evaporate on the Cu grids covered with carbon, with subsequent analysis in a transmission electron microscope (model Fei Tecnai G2 Spirit Twin, Eindhoven, The Netherlands), which operated at 80 kV [80]. The calibration consisted of the diffraction of gold particles, and the distance between two gold nanos (0.24 nm apart) was measured. Previously calibrated nets were used as reference: https://www.agarscientific.com/tem/calibration-standards (accessed on 1 April 2020)

Additionally, the UV–Visible spectrophotometer (Analitikjena Specord S600, Jena, Germany) was used to obtain the absorption spectrum versus the wavelength, as a function of time. The optical absorption spectra of the colloidal nanoparticle solutions were obtained in a wavelength range of 200–800 nm. The baseline was obtained using deionized water as a reference. Colloidal solutions were diluted 1:5 with deionized water prior to analysis [81]. Maintenance, alignment checks, and calibration of the equipment are performed annually. Additionally, an automatic self-recalibration for optimum wavelength accuracy and reproducibility was carried out.

Finally, the SEM was used to determine morphology of the nanoparticles. Samples were previously processed in the Quorum Q150R equipment, in which a carbon fiber coating was placed to further be observed in the SEM (Escan Mira 3, Brno, Czech Republic) [82]. Maintenance, alignment verification, and calibration are performed annually. Calibration was performed with the help of https://www.tedpella.com/calibration_html/SEM_Calibration.aspx (accessed on 1 April 2020) networks.

### 2.7. Growth of Cabbage and Andean Lupin Plants

Lupin seeds, var. I-450 Andino, without infections were phenotypically selected and sown in pots of 22 cm diameter and 20 cm depth, which contained 4 kg of a mixture of crushed soil, coconut pumice, and fiber (1:1:1). Cabbage seedlings, var. Franco, were also sown in other pots, as previously mentioned. The experimental unit was composed of four pots, with six seeds/seedlings each. Once the plants emerged/stablished, a thinning was performed, leaving one cabbage plant or two Adean lupin plants in each pot. The pots were placed in a greenhouse and maintained at a temperature of 12 ± 2 °C night/20 ± 2 °C day (12 h photoperiod), with a relative humidity of 70 ± 10% [83].

The data obtained from the plant’s evaluations were subjected to a one-way analysis of variance (ANOVA) in a completely randomized design with the Infostat Software Version 9, Argentina. Means were compared by using Tukey’s test at 5% level, in order to determine how the treatments were clustered. The *p*-values ≤ 0.05 were considered significant.

### 2.8. Spray of Nanofertilizers to Crops

The nanofertilizers were applied by means of an aqueous solution at the phenological stage of formation of the first 3–4 leaves (cabbage) or 3–4 leaf stage (Andean lupin) [80]. The cultivation period was from May to July 2019, rainfall was 1500 mm, and the average temperature was 15 °C, with a minimum of 12 °C and maximum of 27 °C. For cabbage cultivation, the seedling planting period began on 5 December 2019 and lasted for 66 days in the field trial, while the greenhouse trial began on 19 December of the same year, with a duration of 65 days. Additionally, the completion of each of the phenological cycles was taken as a reference, in order to carry out the pertinent samplings: establishment (31–32 dap (days after planting)); vegetative development (36–38 dap); preformation of the head (50–52 dap); formation of the head (65–66 days).

For Andean lupin cultivation, in the same way as the cabbage crop, the sowing period of the crop, both in the greenhouse and field, began on 26 September 2019 and lasted for 123 days. Additionally, the completion of the phenological cycles was taken as a reference, i.e., flowering (4th), reproductive (5th), and sheathing (6th), to carry out the pertinent samplings, with those being at 92–93, 112–113, and 122–123 dap, respectively. The solution was applied by foliar spray, with 11 mL of each treatment up to the run-off point, so that the nanofertilizers indirectly penetrate the cuticle, or directly through stomata and hydathodes of the plant, after 72 h agronomic data were collected.

Nanoparticle solutions in cabbage with 270 pm of ZnO_MnO-NPs and 540 ppm of ZnO_MnO-NPs were applied and in Andean lupin the solutions with 270 ppm of FeO_ ZnO NPs and 540 ppm of FeO_ZnO-NPs. In cabbage, the response of nanofertilizers was observed during the phenological stages of establishment, vegetative development, preformation, and head formation, while, in Andean lupin, it was evaluated during the stages of vegetative development, flowering, pre-pod, and pod formation.

The effect of nanofertilizers on crop growth was evaluated on the basis of plant height and root length (manual measurement), chlorophyll index (using a spectroradiometer (Spectral Evolution PRS-1100f, Haverhill, MA, USA)), and chlorophyll index (Spectral Evolution PRS-1100f) [71,84,85]. Chlorophyll content was assessed with the Optisciences equipment ( CCM-200 Plus, Hudson, NY, USA), which records information on the wavelengths of 653 and 931 nanometers within the electromagnetic spectrum; measurements were taken at the edge of the leaves of cabbage or Andean lupin, taking into account the size of the leaf and discarding decaying leaves or those with loss of color; six data per plant leaf area (cm^2^/plant) and total biomass (kg/m^2^) were obtained as the sum of the masses of leaves, stems, roots, petiole, and pods, without considering grains, following the given methodologies, respectively [86,87,88].

## 3. Results and Discussion

### 3.1. Characterization of Sugar Cane Bagasse Ash

The proper sugar cane bagasse ash was characterized, and the results are shown in Table 1. Given data are in agreement with similar materials developed on previous studies. The pH was alkaline as found in cachaça (pH 8) and as well as in ash (pH 8.7) from sugarcane [89]. Density of the ash was low (0.5 g/cm^3^), equivalent to densities (0.531g/cm^3^) with a relatively high degree of compressibility and porosity found in a prevoius sugarcane bagasse ash [90]. Additionally, the average size of bottom ash from the muffle was quite similar to samples of a boiler located in Valle del Cauca operating at a temperature of 600 °C of about 79.8 µm [91]. Due to its particle size (1–75 μm), the ash-enriched liquid NPs added as foliar sprays could not contribute to the adsorption of metals (See Table 1).

### 3.2. Characterization of Andean Blue Berry Extract

The antioxidant capacity of Andean blueberry was 52.45%, and the concentration of the polyphenols was 2200 ± 850 (mg GAE/100 g of sample), which was very similar to that published by Murgueitio et al., 2018 [59]. With respect to the size of the polyphenols, it was at the micro-level, as can be seen in Figure 3.

### 3.3. Synthesis of the Nanoparticles

We obtained several reactions during the preparation of the different studied nanoparticles, which were composed of iron, zinc, and manganese. Thus, within the iron nanoparticles (FeO NPs), ferrous sulfate heptahydrate, which is found in a solution of Andean blueberry extract, reacted with the sodium hydroxide solution, which, by adjusting the pH between 8 and 10, dissociates into iron oxide, as shown in reactions 1 to 2.
(1)$$FeSO_{4}+2NaOH\to Fe{\left(OH\right)}_{2}+Na_{2}SO_{4}$$
(2)$$Fe{\left(OH\right)}_{2}\to FeO+H_{2}O$$

In the zinc nanoparticles (ZnO NPs), the reaction was of a double substitution, where the zinc ion joined with the hydroxide ion, and a precipitate was formed (zinc hydroxide); later, it was transformed into zinc oxide, when it was calcined at 450 °C. The hydroxide ions (OH) were eliminated during this calcination, as shown in reactions 3 and 4 [92].
(3)$$Zn\;{\left(O_{2}CCH_{3}\right)}_{2}+2NaOH\;\to Zn{\left(OH\right)}_{2}+2CH_{3}COONa$$
(4)$$Zn{\left(OH\right)}_{2}\to ZnO+H_{2}O$$

A precipitate of manganous hydroxide was obtained within the manganese nanoparticles (MnO NPs); later, the hydroxide was transformed into manganous oxide, by the action on a manganous salt, such as sulfate, as shown in reactions 5 and 6. Manganous oxide is a brown or blackish product, and it is insoluble in water.
(5)$$MnSO_{4}+NaOH\;\to Mn{\left(OH\right)}_{2}+Na_{2}SO_{4}$$
(6)$$Mn{\left(OH\right)}_{2}\to MnO$$

### 3.4. Characterization of NPs

The submicron particle analyzer provided a reference of the diameter of the NPs, ZnO NPs, FeO NPs, and MnO NPs (Figure 4). The NPs had similar diameters, whose values were 9.5 nm ± 1.7 for ZnO NPs, 7.8 nm ± 2.2 for FeO NPs, and 10.1 ± 1.8 nm for MnO NPs, with a concentration of 0.1 mol/L and dilution of 1/100. The ideal concentration of 0.1 mol/L has been used for this type of NPs, since, at a higher concentration, it gives rise to a multi-scattering, in which the particle is dispersed and interacts with others before reaching the detector, eventually intensity [93]. It can be assumed that they were monodisperse and homogeneous solutions, according to the standard deviations of Table 2. The information provided in the graphs indicate a similarity in the diameters of the iron and zinc NPs; however, the manganese NPs have larger sizes (Table 2). They were not cumulative frequency readings—they are individual readings for each type of NPs.

According to [94], the UV–Vis spectra recorded between 300 nm and 550 nm corresponds to ZnO nanoparticles. In Figure 5a, a peak was recorded at the 365 nm wavelength, and it was observed during the reading of the spectra that there were no other peaks, suggesting that the synthesized product could be mostly ZnO NPs. In addition, there was a remarkable variation in the amount of absorbance from 0.45 at 0 h, 0.72 at 1 h, 0.76 at 2 h, and 0.483 at 3 h of its synthesis. Moon et al. [95] observed that chemically synthesized manganese nanoparticles showed a higher absorption at 410 nm, and the appearance of the absorption edge at 360 nm is a clear sign of the formation of these types of nanoparticles [96]. Figure 5b shows the peak of the MnO NPs obtained in the present investigation at 425 nm. The shape of these spectra provides information on the size, shape, size distribution, and surface properties of the particles, as well as the stability over time. The nature of the MnO solution spectrum was asymmetrical, indicating the presence of dispersed particles.

The ZnO NPs present different sizes: 43.40% ranged between 10–15 nm, 12.32% ranged from 5–10 nm, and ranged 0.29% from 30–35 nm; the MnO NPs ranges were 5–10 nm (47.73%), 10–14 nm (30.57%), and 20–25 nm (2.76%) (Table 3; Figure 6). Particle sizes that ranged in the order of 90, 60, and 63 nm were obtained in other studies [75,97,98]. Figure 6b illustrates the MnO NPs, of which 47.73% had diameters between 5–10 nm (Table 2), with these results being similar to previous studies, where the range of MnO nanoparticles were between 6–14 nm, but different to some reported sizes ranging from 65.91 to 110.48 nm [99,100].

The nanoparticles of FeO have a spherical shape, for the most part, and chain-shaped aggregates (Figure 7). In the current study, the size of the nanoparticles presented a diameter less than 70 nm; from these, 64.62% corresponded to a size between 12 and 22 nm (Table 4), which coincided with previously established characteristics [58]. It can be seen that the NPs were found to be covered by a thin film of Andean blueberry fruit. This organic layer prevents them from agglomerating; however, as time goes by, they lose this property and can agglomerate later (see Figure 7a). In the upper rectangle, the NPs were separated; however, in the lower box, they had already started to agglomerate. This agglomeration occurs as their size increases; then, the surface free energy and surface area are reduced. Agglomerates form by various mechanisms, for example due to electrostatic forces between very small (nanoscale) particles, the formation of bridges between particles upon spraying of an additional liquid which may form solid bridges after evaporation of the liquid from a sprayed solution [101].

In Figure 7, it is shown that zinc oxide NPs are similar to those reported by [92], where the size is greater than 100 nm. NPs are shown here as agglomerated clusters; however, when taking a closer look at their shape, similar results to those reported by other researchers are observed [102]. When ZnO NPs are clustered, the system tries to lower their overall surface energy by smoothing out the crystal lattices and reducing the exposed areas and defects. During the growth of the nanoparticles, the surface structure of the particles ends up changing [103,104].

### 3.5. Growth of Cabbage and Andean Lupin Plants

The size of the NPs is considered a significant parameter for studying the uptake in plants, as various barriers that are confined within the plants are in the range of micrometers (µm) to nanometers (nm) [102]. In our study, the diameter of most NPs were less than 12 nm (Table 3 and Table 4 and Figure 5). It has been described that NPs with a size range of 4–100 nm could easily cross the cuticle by disrupting the waxy layer [103]. NPs of 43 nm diameter were able to penetrate the leaves of *Vicia faba* solely through the stomata, whereas particles of 1 µm did not cross at all [104], although stomata containing two guard cells formed a pore about 3–12 μm wide during the opening for gaseous exchange [102]. In our study, foliar spray-applied nanoparticles may penetrate the leaves and be readily translocated to systemic sites.

NPs are transported to different parts of the plant and interact with cellular machinery, thus promoting plant growth, which includes seedling vigor and growth of the roots and shoots [102,104]. Table 5 indicates the effect of NPs sprays on cabbage growth plants, where applications of ZnO–MnO NPs at a concentration of 270 ppm caused increases of 10.3% in root size, 55.1% in dry biomass, 7.1% in chlorophyll content, and 25.6% in leaf area. NPs can improve photosynthesis by interacting with chloroplasts, resulting in an enhanced chlorophyll content [105]. Meanwhile, cabbage plants that were treated at a concentration of 540 ppm yielded an increase of 1.3% in root size and 1.8% in chlorophyll content, compared to the control sprayed with distilled water; however, Tukey’s test, at *p* > 0.05, showed that sprays of 270 and 540 ppm had no significant differences, when compared with the control, both for leaf area and biomass (Table 5). The 1 to 10 nm size NPs also promoted the growth of plants, as evidenced in canola (*Brassica napus*). The highest effect was seen at a concentration range of 400 ppm, with enhanced leaf growth rates and chlorophyll content, as compared to the control [104]. Additionally, it was observed that the nanoparticles had no effect on the height of the treated plants—only 270 ppm increased the leaf area (Table 5). This suggests that the concentration of NPs must be reduced, in order for a more appropriate uptake to promote cabbage growth. Successful foliar uptake, however, depends on additional factors, including species and environmental conditions [105]. New studies are planned by using NPs low dosages between 100–400 ppm in the field and under greenhouse conditions, in order to elucidate a response.

The effect of FeO_ZnO-NPs sprays in Andean Lupin (*L. mutabilis*) are show in Table 6. The application of 270 ppm FeO_ZnO-NPs showed increases of 6% in height and 3.5% in chlorophyll content index; however, root size and leaf area were significantly (*p* > 0.05) higher, when compared to the control. On the other hand, 540 ppm treatment showed no increase in most variables, except for the leaf area, which was significantly (*p* > 0.05) higher than the control. Other studies showed that ZnO nanoparticles at 100 ppm concentrations had a significant effect on the germination of *Lupinus albus* plants, as well as the ability of the seedling tissue cells to uptake zinc oxide under in vitro conditions [106]. Regarding the ZnO NPs, there was also an increase in the shoot length, chlorophyll content, and photosynthesis in the mung bean (*Vigna radiata*) plant with doses of 20 ppm [26]. The better growth of plants, by applying nanofertilizers and incorporating zinc as a growth-stimulating element, compared to the conventional forms, is due to the fast and high uptake level of nanofertilizers, which is seen because of their small size and fast diffusion rates [102]. In our study, the application of NPs at 540 ppm showed no increase in most variables, both on cabbage and Andean lupin plants (Table 5 and Table 6). Other studies showed that plant growth was inhibited when concentrations of NPs were over-applied [26].

The development of nanofertilizers and their impact on agricultural crops have been settled with different types of nanoform fertilizers, as well as their effect on plant growth and physiology. Other studies showed there were also increases in chlorophyll when using 30–60 ppm of NPs in soybean (*Glycine max*) plants [106]. Increased height, in the case of nanofertilizers, is due to their greater potential to supply the plants nutrients than conventional fertilizers [107]. The main agronomic results of zinc oxide NPs were an increase in Zn in the shoots, as well as an initial increase in soil pH, with certain toxicity in radish, vetch, and wheat, and a reduction in wheat growth [94,95]. This suggests that NPs, at doses around or below 250 ppm, may have a better effect on Andean lupin growth. New studies with NPs doses from 50 to 250 ppm are planned to elucidate its effect on Andean lupin growth promotion.

On the other hand, the Andean blueberry extract, used together with the nanoparticles, may provide additional sources that could contribute nutritionally to Andean lupin and cabbage plants. Other studies show that blueberry has secondary metabolites available, such as as metal ions Fe, K, Ca, Mg, Cu, Zn (mg/100 g FW), ascorbic acid (mg/100 g FW) 9.0 ± 2.0, β-carotene (μg/100 g FW) 36.0 ± 6.0, and soluble sugars [108]. In this research, polyphenols could possibly not enter the stomata, due to their micro-level size, which is larger than stomata—the same is likely true with ash. In the study by [109], indicates that secondary metabolites play a critical role in the processes during stress acclimation, so deciphering their relevant responses exchanges the interpretation of the underlying molecular mechanisms may contribute to improved adaptability and efficiency. In this research, Andean lupin and cabbage plants were not subjected to stress; therefore, the NPs with andean blueberry extract applied could hardly modify the concentration of polyphenols.

Nanomaterials are currently being explored as a viable means of improving plant growth and productivity, and some studies suggest employment in the postharvest sector, as well [105]. Regarding sugarcane bagasse, it has been considered that bagasse ash may be an adequate source of micronutrients, such as Fe, Mn, Zn, and Cu [110,111]. It can also be used as soil additive in agriculture, due to its capacity to supply the plants with small amounts of nutrients; therefore, its use in agriculture for crop production will prove more beneficial [112]. Along with positive effect on soil nutrient contents, bagasse ash has also produced an increased yield of wheat crop [113]. However, in edible crops, adverse effects of nanomaterials on human health and the environment have been suggested. In this regard, no laxity in the application ought to be tolerated, and disposal issues ought to be deliberated before commercial use [110].

## 4. Conclusions

Nanoparticles (NPs) ZnO–MnO and FeO–ZnO, enriched with sugarcane bagasse ash and Andean blueberry, were applied to cabbage and Andean lupin plants to increase their growth, compared to that of control plants.

The NPs of Fe, Mn, and Zn were synthesized according to the nutritional importance of these elements, and to potentiate agronomic traits, the individual NPs were joined, forming multicomponent NPs. In addition, the synthesis of NPs used a green and low-cost synthesis method, and no hazardous chemicals were used. The Fe NPs were prepared with Andean blueberry extract (to prevent them from agglomerating) and sugarcane bagasse ash, due to the nutritional contribution, in addition to the fact that it is technically considered waste and can, for this purpose, be revalued. 

The properties, size, distribution, and morphology were determined by DLS, TEM, SEM, and UV–Vis techniques, where we were able to yield that the NPs ranged in size from 5 nm for ZnO NPs to 22 nm for FeO NPs. The modal size of the zinc nanoparticles was between 10 and 15 nm; for manganese, the modal size of the nanoparticles was between 5 and 10 nm, and it was between 12 and 22 nm for the iron nanoparticles.

Furthermore, it was observed that the application of ZnO–MnO and FeO–ZnO NPs that were enriched with sugarcane bagasse ash–Andean blueberry extract in both cabbage and Andean lupin increased the root size, leaf area, and biomass; additionally, it significantly stimulated the chlorophyll content index, as compared to the blank. The phenolic compounds of the Andean blackberry extract were of a macro-level, in order to prevent the NPs from agglomerating, as well as for the conservation of the NPs; therefore, according to the ease of adsorption of the plants, the NPs will enter first. Since they were added through foliar spray, it is feasible that they would easily pass through the pores of the developing plants. According to the size of the NPs, they were easily adsorbed by plant stomata; this was not the case for polyphenols and ashes, whose sizes were larger than that of plant stomata. Last but not least, it has been demonstrated that multicomponent NPs were able to be used as a plant growth enhancer.

## Figures and Tables

**Figure 1 nanomaterials-12-01921-f001:**
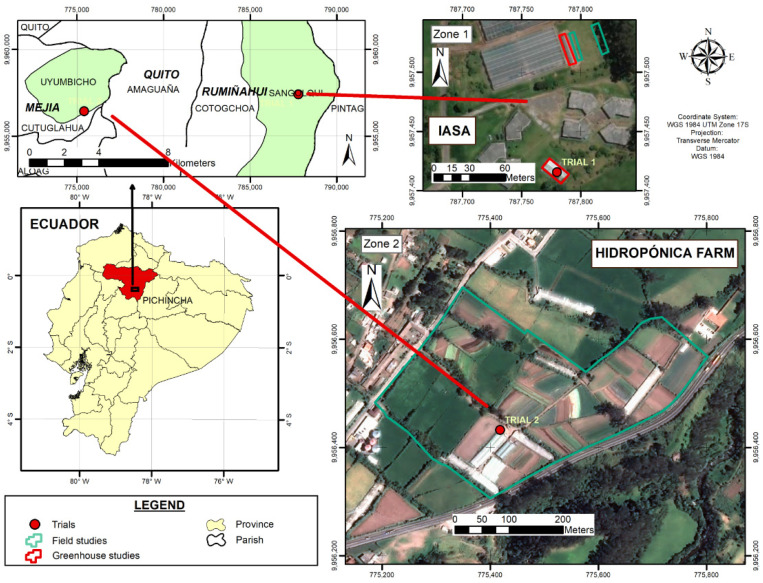
Location of the implemented trials [71].

**Figure 2 nanomaterials-12-01921-f002:**
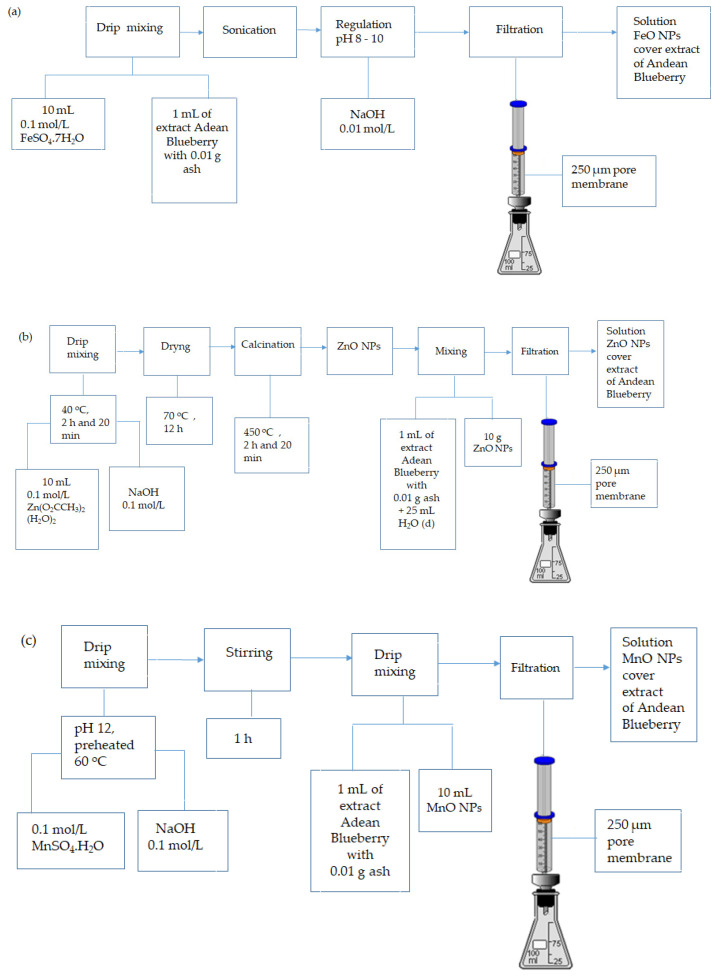
The process diagram for the synthesis of (**a**) FeO, (**b**) ZnO, and (**c**) MgO NPs.

**Figure 3 nanomaterials-12-01921-f003:**
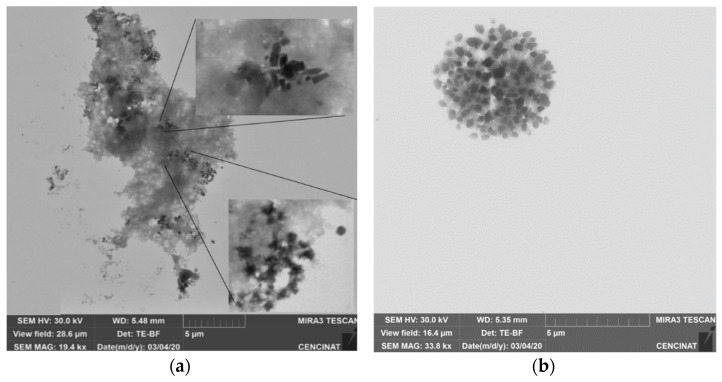
(**a**) FeO NPs agglomerated in the extract, and (**b**) FeO NPs agglomerated by adhesion between particles.

**Figure 4 nanomaterials-12-01921-f004:**
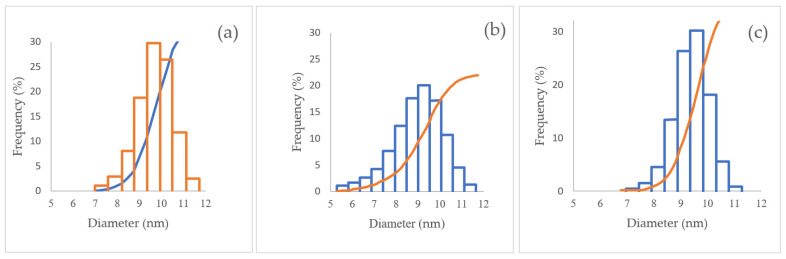
(**a**) Zinc oxide nanoparticles (ZnO NPs), 0.1 mol/L; (**b**) iron oxide nanoparticles (III) (FeO NPs), 0.1 mol/L; (**c**) manganese nanoparticles (MnO NPs), 0.1 mol/L.

**Figure 5 nanomaterials-12-01921-f005:**
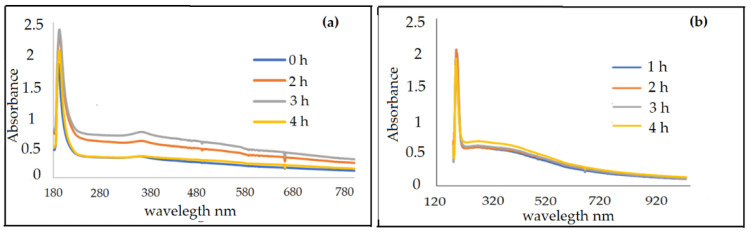
(**a**) UV–Vis spectra of ZnO NPs synthesis, as a function of time. (**b**) UV–Vis spectra of the synthesis of MnO NPs.

**Figure 6 nanomaterials-12-01921-f006:**
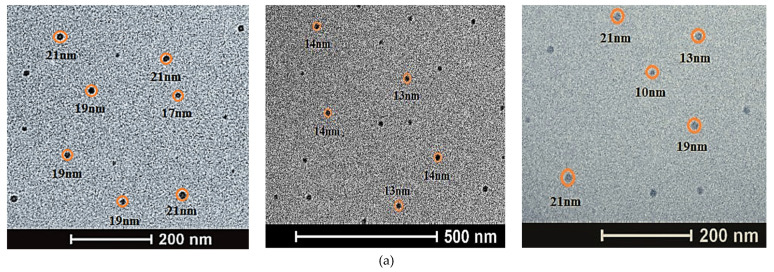
(**a**) The 0.1 mol/L zinc oxide nanoparticles. (**b**) The 0.1 mol/L manganese oxide II nanoparticles.

**Figure 7 nanomaterials-12-01921-f007:**
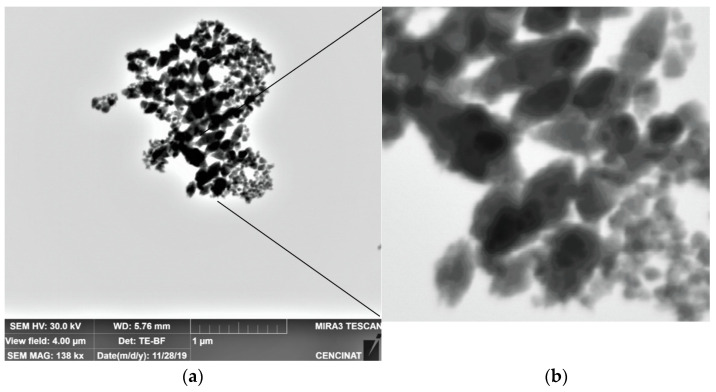
(**a**) SEM; nanoparticles of ZnO 0.1 mol/L, with view field 4.00 µm; (**b**) zoomed image.

**Table 1 nanomaterials-12-01921-t001:** Physicochemical parameters of sugar cane ash.

Parameters	Values
Apparent density	0.5 g/cm^3^
pH	11.2
Electrical conductivity	1427 µs/cm (24.9 °C).
Humidity percentage	0.005%
Fe	5.428 mg/g
Mn	0.567 mg/g
Zn	2.623 mg/g
Granulometric analysis	58.03% ˂ a 75 µm 41.97% 63–1 µm

**Table 2 nanomaterials-12-01921-t002:** Characteristics of the NPs.

	ZnO NPs (0.1 mol/L)	FeO NPs (0.1 mol/L)	MnO NPs (0.1 mol/L)
S.P AREA	6.4779 × 10^6^ (cm^2^/cm^3)^	8.298 × 10^6^ (cm^2^/cm^3)^	6.064 × 10^6^ (cm^2^/cm^3)^
median	9.5 (nm)	7.8 (nm)	10.1 (nm)
mean	9.6 (nm)	7.9 (nm)	10.2 (nm)
Variance	2.9218 (mm^2^)	4.7152 (mm^2^)	3.2688 (mm^2^)
S.D Standard Deviation	1.7 (nm)	2.2 (nm)	1.8 (nm)

**Table 3 nanomaterials-12-01921-t003:** Percentage presence of MnO NPs, in relation to diameter.

Diameter	ZnO NPs 0.1 mol/L (%)	MnO NPs 0.1 mol/L (%)
5–10 nm	12.32	47.73
10–15 nm	43.40	30.57
15–20 nm	32.55	18.93
20–25 nm	10.85	2.76
25–30 nm	0.59	-
30–35 nm	0.29	-

**Table 4 nanomaterials-12-01921-t004:** Percentage presence of 0.1 mol/L FeO NPs, in relation to diameter.

Diameter (nm)	FeO NPs 0.1 mol/L (%)
12–22	64.62
22–32	9.23
32–42	12.31
42–52	3.08
52–62	7.69
62–70	3.08

**Table 5 nanomaterials-12-01921-t005:** Effect of ZnO–MnO NPs sprays on growth of cabbage (*Brassica oleracea var. capitata*) cultivar Franco.

Variables	Treatments		
	Control	ZnO_MnO-NPs 270 ppm	ZnO_MnO-NPs 540 ppm
Height (m)	0.2767	0.2100	0.2133
Root (m)	0.1300	0.1433	0.1317
Biomass (kg/m^2^)	0.7900	1.2250	0.7440
Chlorophyll Content Index	0.1737	0.1860	0.1769
Leaf area (cm^2^/plant)	0.0380	0.0477	0.0304

**Table 6 nanomaterials-12-01921-t006:** Effect of FeO_ZnO-NPs on growth of Andean lupin (*Lupinus mutabilis* Sweet), var. I-450 Andino.

Variables	Treatments		
	Control (without nanoparticles)	FeO_ZnO-NPs 270 ppm	FeO_ZnO-NPs 540 ppm
Plant Height (m)	1.56	1.66	1.56
Root size (m)	0.47	0.56	0.47
Biomass (g/m^2^)	0.11	0.09	0.09
Chlorophyll content Index	4.20	4.35	4.19
Leaf area	0.240	0.964	1.019

## Data Availability

Not applicable.

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
