# Peer review of "Synthesis of Iron, Zinc, and Manganese Nanofertilizers, Using Andean Blueberry Extract, and Their Effect in the Growth of Cabbage and Lupin Plants"

_nanomaterials, 2022, doi:10.3390/nano12111921_

Round 1

Reviewer 1 Report

The present manuscript deals with the growth promoting effect of spraying three type of nanoparticles during plant growth in two commercial crops. The reported findings are interesting. The authors are requested to deal with the following issues within the manuscript (rather than the response letter).

(1) no need to provide the latin names in the title, please remove

(2) Abstract:

(a) not clear that the ZnO_MnO was tested in cabbage, and the FeO_ZnO was tested in lupin; (b) why highlight the use of ash powder, since the effect of nano-fertilizers is finally evaluated; (c) results on the effect of FeO_ZnO on lupin are only documented! What about the effects of ZnO_MnO on cabbage growth? and

(d) mention some results about the particle size of nanoparticles (see line 390), since this is important for mobility purposes

(e) not clear that Zn is always included in the nanoparticles, and in this way you actually compare the effect of Mn with Fe

(f) what is your conclusion based on the dose-response?

(3) Line 95: not clear that Zn is always included in the nanoparticles, and in this way you actually compare the effect of Mn with Fe

(4) Lines 116–118: indeed, among antioxidants, phenolic compounds are of particular interest due to their strong potential to reduce stress events both at cellular and organismal levels (see Pappi et al., 2021 Horticulturae 7, 448). But then, how do you know that the noted effects are associated with the nanoparticles under study, and not to the antioxidants of the extract?

(5) for how long was the cultivation period, and which period of the year did it take place?

(6) how did you measure root size? Is this merely root length?

(7) Tables 5 and 6: the units are missing in biomass, and leaf area.

(8) You need to stress that spraying nanoparticles in environmental friendly, and cost-effective as compared to soil applications. In real world situations, spraying the solution is undoubtedly more practical and feasible than methods targeting uptake through the stem (Ahmadi-Majd et al. 2021 J. Hortic. Sci. Biotechnol. 10.1080/14620316.2021.1993755). In this case, notably, the solution volume required to spray is considerably less than the one needed for stem uptake. This sizeable difference sets spraying not only more inexpensive for the agricultural industry, but also minimises the environmental impact (Ahmadi-Majd et al. 2021 J. Hortic. Sci. Biotechnol. 10.1080/14620316.2021.1993755).

(9) Application of nanoparticles not only through the root, but also via foliar spray has been shown to be feasible in developing plants. Foliar spray-applied ones effectively penetrate into the leaves and readily translocate to systemic sites (Ahmadi-Majd et al. 2022 Chem. Biol. Technol. 9, 15). Successful foliar uptake, however, depends on additional factors, including species and environmental conditions (Ahmadi-Majd et al. 2022 Chem. Biol. Technol. 9, 15). One of these factors is particle size, and this is why it deserves to be discussed.

(10) Line 384: Nanomaterials are currently explored as a viable means of improving plant growth and productivity, and some studies suggest employment in the postharvest sector too (Ahmadi-Majd et al. 2021 J. Hortic. Sci. Biotechnol. 10.1080/14620316.2021.1993755). In edible crops, adverse effects of nanomaterials on human health and environment have been suggested. In this regard, no laxity in application ought to be tolerated, and disposal issues ought to be deliberated before commercial use (Ahmadi-Majd et al. 2022 Chem. Biol. Technol. 9, 15).

(11) Authors need to fill the contribution section (line 399)

(12) how do the authors explain that nanoparticles promoted growth in one species, whereas no effect is noted in the other one? What this finding suggests? Is it possible that in the non-affected species, a different concentration may have worked?

(13) I am not familiar with the methods used to assess chlorophyll content. Can you please describe more in detail? What are the units in these measurements?

(14) Lines 116/ 365: latin name in italics

(15) line 205: 3 materials were used (thus why 4 units in parenthesis?)

(16) who is the intended audience profiting from the obtained results? Farmers, scientists, students? 

(17) why did you select these two cultivars?  

Author Response

Reply report based on: Comments and Suggestions for Authors #1

The present manuscript deals with the growth promoting effect of spraying three type of nanoparticles during plant growth in two commercial crops. The reported findings are interesting. The authors are requested to deal with the following issues within the manuscript (rather than the response letter).

  • no need to provide the latin names in the title, please remove

Answer: we changed as recommended. Thanks for the suggestion. Its now “Synthesis of iron, zinc and manganese nanofertilizers and their effect in the growth of cabbage and lupin plants”

 (2) Abstract:

(a) not clear that the ZnO_MnO was tested in cabbage, and the FeO_ZnO was tested in lupin; (b) why highlight the use of ash powder, since the effect of nano-fertilizers is finally evaluated; (c) results on the effect of FeO_ZnO on lupin are only documented! What about the effects of ZnO_MnO on cabbage growth? And

  1. a) and c) answer: Abstract: The predominant aim of the current study was to synthesize nanofertilizer nanoparticles (NPs) ZnO_MnO and NPs FeO_ZnO, using Andean blueberry extract and determine the effect of NPs in growth promotion of cabbage (Brassica oleracea var. capitata) and lupin (Lupinus mutabilis Sweet) crops. The nanoparticles were analyzed by visible spectrophotometry, size distribution (DLS), scanning electron microscopy (SEM) and transmission electron microscopy (TEM).Solutions of nanoparticle concentrations were applied to cabbage with 270 pm and 540 ppm of NPsZnO_MnO and solutions with 270 ppm and 540 ppm of NPsFeO_ZnO were applied to lupine, Zinc was used in both plants to take advantage of its fungicidal potential and beneficial properties on plant growth. Foliar NPs sprays were applied on the phenological stage of vegetative growth of cabbage or lupin plants grown under greenhouse conditions. The solution volume required to foliar spray was considerably less than the usually reported for stem uptake. The diameter of the NPs was 9.5nm ZnO, 7.8nm FeO and 10.5nm MnO, which facilitates the adsorption of NPs by the stomata of plants that have a size between 0.6-3.5 micrometers. In lupin the treatment with 270 ppm of iron and zinc indicated an increase of 6% in height, 19% in root size, 3.5% in Chlorophyll Content Index, and 300% in leaf area, while the treatment with 540 ppm of iron and zinc yielded no apparent increase in any variable. In cabagge the NPsZnO_MnO on Cabbage indicate that at a concentration of 270 ppm had an increase of 10.3% in root size, 55.1% in dry biomass, 7.1% in chlorophyll content and 25.6% in leaf area. For its part, cabbage plants treated at a concentration of 540 ppm produced an increase of 1.3% in root size and 1.8% in chlorophyll content, compared to the control sprayed with water distilled. Nanofertilizer applications at 270 ppm  demonstrated an improvement in growth.

Keywords: nanofertilizers, nanoparticles, lupin, cabbage, Andean blueberry extract.

  1. B) Answer: Indeed, it was demonstrated that the ash does not contribute significantly to the growth of lupine and cabbage . LINE 507-508. however in this research polyphenols possibly could not enter stomata due to their micro level size, larger than stomata , likewise with ash.

(d) mention some results about the particle size of nanoparticles (see line 390), since this is important for mobility purposes.

Abstract: The predominant aim of the current study was to synthesize nanofertilizer nanoparticles (NPs) ZnO_MnO and NPs FeO_ZnO, using Andean blueberry extract  and determine the effect of NPs in growth promotion of cabbage (Brassica oleracea var. capitata) and lupin (Lupinus mutabilis Sweet) crops. The nanoparticles were analyzed by visible spectrophotometry, size distribution (DLS), scanning electron microscopy (SEM) and transmission electron microscopy (TEM).Solutions of nanoparticle concentrations were applied to cabbage with 270 pm and 540 ppm of NPsZnO_MnO and solutions with 270 ppm and 540 ppm of NPsFeO_ZnO were applied to lupine, Zinc was used in both plants to take advantage of its fungicidal potential and beneficial properties on plant growth.  Foliar NPs sprays were applied on the phenological stage of vegetative growth of cabbage or lupin plants grown under greenhouse conditions. The solution volume required to foliar spray was considerably less than the usually reported for stem uptake. The diameter of the NPs was 9.5nm ZnO, 7.8nm FeO and 10.5nm MnO, which facilitates the adsorption of NPs by the stomata of plants that have a size between 0.6-3.5 micrometers. In lupin the treatment with 270 ppm of iron and zinc indicated an increase of 6% in height, 19% in root size, 3.5% in Chlorophyll Content Index, and 300% in leaf area, while the treatment with 540 ppm of iron and zinc yielded no apparent increase in any variable. In cabagge the NPsZnO_MnO on Cabbage indicate that at a concentration of 270 ppm had an increase of 10.3% in root size, 55.1% in dry biomass, 7.1% in chlorophyll content and 25.6% in leaf area. For its part, cabbage plants treated at a concentration of 540 ppm produced an increase of 1.3% in root size and 1.8% in chlorophyll content, compared to the control sprayed with water distilled. Nanofertilizer applications at 270 ppm  demonstrated an improvement in growth.

 (e)  not clear that Zn is always included in the nanoparticles, and in this way you actually compare the effect of Mn with Fe.

Answer:  LINE 86-98

The effectiveness of ZnO NPs has been proven individually by several researches, in the application of different plants, it is known that ZnO NPs reduce the presence of diseases by their antifungal activity against Penicillium expansum, Botrytis cinerea, Aspergillus flavus, Aspergillus niger, Aspergillus fumigatus, Fusarium culmorum and Fusarium oxysporum, Botrytis cinerea, Aspergillus flavus, Aspergillus niger, Aspergillus fumigatus, Fusarium culmorum, and Fusarium oxysporum [1] [2] [3] and antimicrobials for crop protection [4]. Their antifungal effect is partly because they cause malformation of hyphae leading to fungal death [5]. According to [6]; [7], they improve plant growth and fruit quality by increasing sugar concentration [8], [9]; [10] point out that the application of metallic NPs such as zinc , have shown significant effects on seed germination and plant growth. Other reports indicate a phytotoxic effect on different cultivated plants. Iron plays a significant role in enhancing plants’ yield, biomass, and plants’ growth by providing essential nutrients [11].

Corresponding references:

[1]

P. Rajiv, S. Rajeshwari y R. Venckatesh, «Bio-Fabrication of zinc oxide nanoparticles using leaf extract of Parthenium hysterophorus L. and its size-dependent antifungal activity against plant fungal pathogens.,» Spectrochimica Acta Part A: Molecular and Biomolecular Spectroscopy., pp. 384-387, 2013.

[2]

L. He, Y. Liu, A. Mustapha y M. Lin, «Antifungal activity of zinc oxide nanoparticles against Botrytis cinerea and Penicillium expansum,» Microbiological Research, vol. 166, nº 3, pp. 207-215, Marzo 2011.

[3]

G. D. Kumar, N. Natarajan y S. Nakkeeran, «Antifungal activity of nanofungicide Trifloxystrobin 25% + Tebuconazole 50% against Macrophomina phaseolina,» African Journal of Microbiology Research, vol. 10, nº 4, pp. 100-105, 2016.

[4]

S. Shende, A. P. Inlge, A. Gade y M. Rai, «Green synthesis of copper nanoparticles by Citrus medica Linn. (Idilimbu) juice and its antimicrobial activity,» World Journal of Microbiology and Biotechnology, vol. 31, nº 6, pp. 865-873, Junio 2015.

[5]

A. Servin, W. Elmer, R. De la Torre-Roche, H. Hamdi, J. C. White, P. Bindraban y C. Dimkpa, «A review of the use of engineered nanomaterials to suppress plant disease and enhance crop yield,» Journal of Nanoparticle Research, vol. 17, nº 2, 2015.

[6]

R. Liu y R. Lal, «Potentials of engineered nanoparticles as fertilizers for increasing agronomic productions,» Science of The Total Environment, vol. 514, pp. 131-139, Mayo 2015.

[7]

L. Zhao, Y. Sun, J. A. Hernandez-Viezca, A. D. Servin, J. Hong, G. Niu, J. R. Peralta-Videa, M. Duarte-Gardea y J. L. Gardea-Torresdey, «Influence of CeO2 and ZnO Nanoparticles on Cucumber Physiological Markers and Bioaccumulation of Ce and Zn: A Life Cycle Study,» Journal of Agricultural and Food Chemistry, vol. 61, nº 49, pp. 11945-11951, 11 Diciembre 2013.

[8]

A. C. Pandey, S. S. Sanjay y R. S. Yadav, «Application of ZnO nanoparticles in influencing the growth rate of Cicer arietinum,» Journal of Experimental Nanoscience, vol. 5, nº 6, pp. 488-497, Diciembre 2010.

[9]

T. N. Prasad, P. Sudhakar, Y. Sreenivasulu, P. Latha, V. Munaswamy, K. R. Reddy, T. S. Sreeprasad, P. R. Sajanlal y T. Pradeep, «EFFECT OF NANOSCALE ZINC OXIDE PARTICLES ON THE GERMINATION, GROWTH AND YIELD OF PEANUT,» Journal of Plant Nutrition, vol. 35, nº 6, pp. 905-927, Abril 2012.

[10]

U. Burman, M. Saini y P. Kumar, «Effect of zinc oxide nanoparticles on growth and antioxidant system of chickpea seedlings,» Toxicological & Environmental Chemistry, vol. 95, nº 4, pp. 605-612, Abril 2013.

[11]

A. Konate, X. He, Z. Zhang, Y. Ma, P. Zhang, G. Alugongo, .. M y Y. Rui, «Magnetic (Fe3O4) nanoparticles reduce heavy metals uptake and mitigate their toxicity in wheat seedling,» Sustainability, vol. 9, nº 5, p. 790.

[12]

I. Sinde-Gonzalez, J. Gómez López, A. Tapia-Navarro, E. Murgueitio, C. Falconí, F. Benítez y T. Toulkeridis, «Determining the Effects of Nanonutrient Application in Cabbage (Brassica oleracea var. capitate L.) Using Spectrometry and Biomass Estimation with UAV.,» Agronomy, vol. 12, nº 1, p. 81, 2021.

[13]

M. Gutiérrez, J. Escalante, M. Rodríguez y M. Reynolds, «Índices de reflectancia y rendimiento del frijol con aplicaciones de nitrógeno.,» Terra Latinoamericana, vol. 22, nº 4, pp. 409-416, 2004.

[14]

A. Gitelson, A. Viña, T. Arkebauer, D. Rundquist, G. Keydan y B. Leavitt, «Remote estimation of leaf area index and green leaf biomass in maize canopies.,» Geophysical research letters, vol. 30, nº 5, 2003.

[15]

V. Yánez-Mendizábal y C. Falconí, «Bacillus subtilis CtpxS2-1 induces systemic resistance against anthracnose in Andean lupin by lipopeptide production.,» Biotechnol Lett, vol. 43, pp. 719-729, 2021.

[16]

M. Barrios, A. Buján, S. Debelis, A. Sokolowski, H. Rodríguez, ..... y M. Gagey, «Relación Biomasa de Raíz/Biomasa Total de Soja (Glycine Max) en dos Sistemas de Labranza.,» Terra Latinoamericana, vol. 32, nº 3, pp. 221-230., 2014.

[17]

V. Yánez-Mendizábal y C. Falconí, «Efficacy of Bacillus spp. to biocontrol of anthracnose and enhance plant growth on Andean lupin seeds by lipopeptide production.,» Biological control, vol. 122, pp. 62-75, 2018.

 (f) what is your conclusion based on the dose-response?

Answer: As it is an experimental work, the doses that were tested of 270 ppm and 540 ppm that were used resulted in some unfavorable cases due to the inhibition in the following aspects: in the cabagge in its Height (m) and in the Leaf area at 540 ppm. In the case of the lupine in the Plant Height (m), Biomass.

(3) Line 95: not clear that Zn is always included in the nanoparticles, and in this way you actually compare the effect of Mn with Fe

Answer: Previously replied at (2)

(4) Lines 116–118: indeed, among antioxidants, phenolic compounds are of particular interest due to their strong potential to reduce stress events both at cellular and organismal levels (see Pappi et al., 2021 Horticulturae 7, 448). But then, how do you know that the noted effects are associated with the nanoparticles under study, and not to the antioxidants of the extract?.

Answer: LINE 507. In the study by  Pappy et al 2021, indicates that secondary metabolites play a critical role in the processes during stress acclimation, so deciphering their relevant responses exchanges the interpretation of the underlying molecular mechanisms that may contribute to improved adaptability and efficiency. In this research lupin and cabbage plants were not subjected to stress therefore the NPs with andean blueberry extract applied could hardly modify the concentration of polyphenols .

Corresponding reference : P. Pappi, N. Nikoloudakis, D. Fanourakis, A. Zambounis, C. Delis and G. Tsaniklidis, "Differential triggering of the phenylpropanoid biosynthetic pathway key genes transcription upon cold stress and viral infection in tomato leaves.," Horticulturae, vol. 7, no. 11, p. 448, 2021.  

(5) for how long was the cultivation period, and which period of the year did it take place?

Answer: Line 313 -325. Cultivation period was from May to July 2019, during the dry season.

Cabbage cultivation:

The sowing period of the crop began on December 5, 2019 and lasted for 66 days in the field trial, while the greenhouse trial began on December 19 of the same year with a duration of 65 days. Additionally, the completion of each of the phenological cycles was taken as a reference to carry out the pertinent samplings, these being: 1 Establishment. 31-32 dap (days after sowing), 2 Vegetative development. 36-38 dap, 3 preformation of the head. 50-52 daps and, 4 Formation of the head. 65-66 days

dds= days after sowing

Lupine cultivation:

In the same way as the cabbage crop, the sowing period of the crop, both in the greenhouse and in the field, began on September 26, 2019 and lasted for 123 days. Also, the completion of the phenological cycles was taken as a reference; Flowering (4th), Reproductive (5th) and Sheathing (6th) to carry out the pertinent samplings, these being 92-93 DAS, 112-113 DAS and 122-123 DAS, respectively.

(6) how did you measure root size? Is this merely root length?

Answer: The values of morphological characteristics of the root part of the plants were obtained by a conventional mechanical method, taking into account the maximum length of the roots. This measurement was made 5 weeks after the product had been applied. García & Watson (2001), obtained the morphological characteristics of the aerial and radical part of the maize plant by means of the mechanical method, approximately 10 weeks after sowing.

Corresponding reference: García, M., Watson, C., & Salcedo, F. (2001). Evaluación de métodos para determinar resistencia al Acame de raíces en maíz dulce (Zea mays L.). Bioagro13(1), 22-31.

 (7) Tables 5 and 6: the units are missing in biomass, and leaf area.

Answer: Now included in tables 5 and 6.

Table 5. Effect of NPsZnO_MnO enriched with sugarcane bagasse ash sprays on cabbage (Brassica oleracea var. capitata)

Variables

Treatments

Control

NPsZnO_MnO

270ppm

NPsZnO_MnO

540ppm

Height(m)

0.2767

0.2100

0.2133

Root (m)

0.1300

0.1433

0.1317

Biomass (kg/m2)

0.7900

1.2250

0.7440

Chlorophyll Content Index

0.1737

0.1860

0.1769

Leaf area (cm2/plant)

0.0380

0.0477

0.0304

(8) You need to stress that spraying nanoparticles in environmental friendly, and cost-effective as compared to soil applications. In real world situations, spraying the solution is undoubtedly more practical and feasible than methods targeting uptake through the stem (Ahmadi-Majd et al. 2021 J. Hortic. Sci. Biotechnol. 10.1080/14620316.2021.1993755). In this case, notably, the solution volume required to spray is considerably less than the one needed for stem uptake. This sizeable difference sets spraying not only more inexpensive for the agricultural industry, but also minimises the environmental impact (Ahmadi-Majd et al. 2021 J. Hortic. Sci. Biotechnol. 10.1080/14620316.2021.1993755).

Answer: Lines 326-329. The solution was applied by foliar spray, 11 ml of each treatment up to the run-off point, so that the nanofertilizers penetrate indirectly through the cuticle, or directly through stomata and hydathodes of the plant.

(9) Application of nanoparticles not only through the root, but also via foliar spray has been shown to be feasible in developing plants. Foliar spray-applied ones effectively penetrate into the leaves and readily translocate to systemic sites (Ahmadi-Majd et al. 2022 Chem. Biol. Technol. 9, 15). Successful foliar uptake, however, depends on additional factors, including species and environmental conditions (Ahmadi-Majd et al. 2022 Chem. Biol. Technol. 9, 15). One of these factors is particle size, and this is why it deserves to be discussed.

Answer: You are right. It was included in the discussion.

(10) Line 384: Nanomaterials are currently explored as a viable means of improving plant growth and productivity, and some studies suggest employment in the postharvest sector too (Ahmadi-Majd et al. 2021 J. Hortic. Sci. Biotechnol. 10.1080/14620316.2021.1993755). In edible crops, adverse effects of nanomaterials on human health and environment have been suggested. In this regard, no laxity in application ought to be tolerated, and disposal issues ought to be deliberated before commercial use (Ahmadi-Majd et al. 2022 Chem. Biol. Technol. 9, 15).

Answer: You are right. It was included in discussion.

(11) Authors need to fill the contribution section (line 399)

Answer: Done

(12) how do the authors explain that nanoparticles promoted growth in one species, whereas no effect is noted in the other one? What this finding suggests? Is it possible that in the non-affected species, a different concentration may have worked?

Answer: Successful foliar uptake, however, depends on additional factors, including species and environmental conditions (Ahmadi-Majd et al. 2022). Included in discussion.

(13) I am not familiar with the methods used to assess chlorophyll content. Can you please describe more in detail? What are the units in these measurements?

Answer: Added in lines 336-346.

The effect of nanofertilizers on crop growth was evaluated on the basis of plant height and root length (manual measurement), chlorophyll index using a spectroradiometer (Spectral Evolution PRS-1100f) and chlorophyll index (Spectral Evolution PRS-1100f) [66, 79, 80]. Chlorophyll content was as seed by the Optisciences CCM-200 Plus equipment that record information on the wavelengths of 653 and 931 nanometers within the electromagnetic spectrum; measurements were made at the edge of the leaves of cabbage or lupin, taking into account the size of the leaf and discarding decaying leaves or with loss of color, six data per plant Leaf area (cm2/plant) and total biomass (kg/m2) was obtained as the sum of the masses of leaves, stems, roots, petiole and pods without considering grains, following the given methodologies, respectively [81, 82, 83].

(14) Lines 116/ 365: latin name in italics  ok

Line 189   Vaccinium floribundum

Line 510      Glycine max

(15) line 205: 3 materials were used (thus why 4 units in parenthesis?)

Line 205        (1:1:1)

(16) who is the intended audience profiting from the obtained results? Farmers, scientists, students?.

Answer: Thanks for your concern. Mainly the Ecuadorian population that consumes lupine and cabagge, due to the importance of improving the nutritional contribution and phenological aspects of these plants and later as a scientific contribution to Farmers, scientists, students. It has also obviously a worldwide interest, beside the national one

(17) why did you select these two cultivars?  

Answer: Lines 153-160.

Thanks a lot for you kind comments, questions and recommendations, which have led to obtain a much clearer and more fluid text in our manuscript. Your input has helped us a lot and we are very grateful for your concerns. Now, due to your support, the manuscript improved considerably.

Reviewer 2 Report

The authors' point is that the application of nanoscale fertilizer is effective for plant growth.  I was wondering how nanoscale substances are taken to plants and used. Normally, it would be faster to work in liquid form. I would like you to describe how nano-fertilizer is taken in and then used by plants.

L101

Please clarify Zone 1 and Zone 2 in the map.

L119
It says its level of maturity, but does the effect differ depending on the level?

L219

pmâž¡ppm

L242

Table 1
Why is the unit of Fe, Mn, Zn in the incinerator ash the concentration?
Isn't it displayed as mg / g if it is in a solid? ..
How were the electrical conductivity and pH measured?
Do you  suspended ash in water?

L283

Is it okay for the curves in the figure to be frequency? Isn't it cumulative frequency?

L302

What is the time in Figure 4b?

Tables 5 and 6

I think you need a unit of biomass

Author Response

Reply report based on: Comments and Suggestions for Authors #2

The authors' point is that the application of nanoscale fertilizer is effective for plant growth.  I was wondering how nanoscale substances are taken to plants and used. Normally, it would be faster to work in liquid form. I would like you to describe how nano-fertilizer is taken in and then used by plants.

Answer: The NPs that were applied to the plants are liquid.

L101

Please clarify Zone 1 and Zone 2 in the map.

Answer: LINE 184 

L119
It says its level of maturity, but does the effect differ depending on the level?

Answer: Only reddish-black color fruit used, because other compounds are present in green or overripe fruit. Done.

L219

pmâž¡ppm 

OK, done

L242

Table 1
Why is the unit of Fe, Mn, Zn in the incinerator ash the concentration?

Answer: 3.1 Characterization of sugar cane bagasse ash Table 1. Physicochemical parameters of sugar cane ash.. LINE 361 mg/g  .Done

Isn't it displayed as mg / g if it is in a solid?

Answer: Yes indeed our mistake. Done . Line  361

How were the electrical conductivity and pH measured?

Answer:  LINE 208. 2.4 Characterization of sugar cane bagasse ash .

Do you  suspended ash in water?

Answer:  LINE 208. 2.4 Characterization of sugar cane bagasse ash .

How were the electrical conductivity and pH measured?

Answer: LINE 208. 2.4 Characterization of sugar cane bagasse ash .

The ash was suspended in 50 ml of distilled water, stirred and this solution was used to measure pH and electrical conductivity using a HACH multiparameter meter, model HQ30 d  [68].To determine the hydrogen potential, it was calibrated with buffer solutions of 4, 7, 9 pH. Resolution: Selectable between 0.001 to 0.1 pH. pH Electrode calibration: 0.002. pH Measurement: 0 to 14 pH. For electrical conductivity it was calibrated with a standard solution of 1413µS/cm.Conductivity Measurement Range: 0.01 μS/cm to 200 mS/cm . Conductivity resolution: 0.01 μS/cm with 2 digits. Conductivity Accuracy: ± 0.5 % from (1μS/cm - 200 mS/cm).Conductivity and pH determined following methods 4500-H+B and 2510B, respectively of Standard Methods 18th Edition using the HQ30d Multiparameter Meter, Hach. Bulk density was determined using the given protocol [69, 70]. Grain size and moisture content analyses were performed following the methodology established by Manals et al. [68]. Metals (Fe, Mn and Zn) were determined by flame atomic absorption using the AAnalysit 800 Atomic Absorption Spectrometer Pelkin Elmer, following the methodology detailed in the Standard Methods 18th Edition, using method 3111-B [69]. Iron, manganese and zinc were determined by AA with AAnalyst 800 PE equipment by method 3111 B ( direct air-acetylene flame method ), Fe ( %Rec. 101%, RDS : 2.35) Mn ( %Rec. 99%, RDS 1.02); Zn (% Rec 104%, RDS 2.35). RSD is the Relative Deviation

Do you  suspended ash in water?.

Answer: LINE 208 . If the ash was suspended in distilled water and stirred to later take the pH and electrical conductivity readings

Is it okay for the curves in the figure to be frequency? Isn't it cumulative frequency?

Answer: LINE 404 . 3.3 Characterization of NPs.  They are not cumulative frequency readings, they are individual readings for each type of NPs. They are cumulative when readings overlap.

L302

What is the time in Figure 4b?.

Answer: According to the colors of the graph you can see the times of 0, 2, 3 and 4 hours.

Tables 5 and 6

I think you need a unit of biomass

Indeed, therefore corrected already

Thanks a lot for you kind comments, questions and recommendations, which have led to obtain a much clearer and more fluid text in our manuscript. Your input has helped us a lot and we are very grateful for your concerns. Now, due to your support, the manuscript improved considerably.

Reviewer 3 Report

The manuscript has an interesting concept and it has novelty. Unfortunately, these strong points are wasted because of the agronomic part that has no clear experimental design. There are other specific comments below that may be helpful to improve the document.

**Title: The authors do not mention the use of the blueberry extract, the most important point in the experiment because, without this extract, the nanoparticles would not be formed. Please, add this information to the title.

**Abstract: The authors do not mention the use of the blueberry extract, the most important point in the experiment because, without this extract, the nanoparticles would not be formed. Please, add this information to the abstract.

**Keywords: Please, add “blueberry extract” as a keyword.

**Introduction:

*The current flow is:

-Food scarcity and need to improve yields > Problems using conventional chemicals > The use of nanofertilizers and examples of success and methods to synthesize them > The use of byproducts to produce nanoparticles and conventional uses of sugarcane bagasse ash and the advantages of nanoformulations for soil remediation > The objective of this research

The flow is confusing and the authors mix different ideas in the same paragraph. I want to propose the following flow to improve the introduction:

-Food scarcity and need to improve yields > Problems using conventional fertilizers > How inadequate fertilization reduces the availability of micronutrients > Problems caused by deficiency of micronutrients, focusing on zinc, iron, and manganese > Use of nanofertilizers as a way to solve the environmental problems > Methods to produce nanofertilizers, including agricultural byproducts > Economic importance of the sugarcane bagasse ash > Objective of this work

-The use of pesticides (second paragraph) is out of the scope of the paragraph and I recommend removing this part of the sentence.

-Please, add a paragraph after the second paragraph explaining how alkaline soils reduce the absorption of zinc and another paragraph later describing problems caused by the deficiency of micronutrients

-N, P, K, S, Ca, and Mg are not micronutrients. Please, fix it.

**Material and Methods

-Please, use “L” for a liter and other units, like “mL”.

*Study area

-I recommend adding the year and months of the experience if the authors agree. It is not the most important information, but it is interesting to allow readers to check how was the rain and temperature during the experiment.

*Andean blueberry extract

-There is no information in the title or introduction about this

*Synthesis of nanoparticles (NPs)

-I understand how the nanoparticles were prepared using blueberry extract. I cannot understand how much ash was mixed with the NP suspension. Did the authors calculate the concentration of nanoparticles and mixed a volume of 1000 nanoparticles with 1 g or 1 mg of ash? Please, rewrite this part of the document to make it clear.

-Please, explain the reasoning to create mixed nanoparticles instead of using them individually.

*Growth of cabbage and lupin plants

-The authors write that the potting substrate is composed of soil, crushed pumice, and coconut fiber. Nevertheless, the ratio is 1:1:1:1, implying that there are 4 components. Please, rewrite this and explain better the substrate.

*Spray of nanofertilizers on the crops

-What do the authors mean by “72 hours before each sampling”? What type of sampling or measurement was performed? It is not possible to understand what the authors evaluated in the experiment based on this description.

*Statistical analysis

-What type of statistical analysis was performed?

**Results and discussion

*Synthesis of the nanoparticles

-Please, write “Cl” for chlorite, not “CL”.

*Characterization of NPs

-The authors successfully produced nanoparticles with the blueberry extract and the salts. The nanoparticles have different morphologies and size distributions. Unfortunately, there are a few problems. There is no explanation of why the authors decided to assemble ZnO and MnO nanoparticles and ZnO and FeO nanoparticles. The authors did not test the nanoparticles without ash particles. They do not explain the reasoning to test the fertilizer with the ash particles. What type of statistical analysis was performed to determine statistical differences between the treatments? Why did the authors use a set of treatments for one crop, and another set of treatments for another crop? There are so many unclear points in the experiment that the best that the authors can explain is that there is evidence that nanofertilizers can improve agronomical characteristics for cultivars of those crops, and other experiments must be performed with proper statistical planning.

-Figure 3: Please, keep the x-axis identical for all graphs

**Conclusions

-The conclusions are based on the results, but I do not agree that the nanoparticles are “improving agronomic yields significantly” because yield is an agronomic characteristic not evaluated in the experiment.

**Authors contributions: Please, replace X.X. and Y.Y. with the name of the authors who worked in each one of the elements of this section.

**Funding: OK

**Data Availability Statement: OK

** Acknowledgments: OK

** Conflicts of Interest: OK

**References: OK

Author Response

Reply report based on: Comments and Suggestions for Authors #3

The manuscript has an interesting concept and it has novelty. Unfortunately, these strong points are wasted because of the agronomic part that has no clear experimental design. There are other specific comments below that may be helpful to improve the document.

**Title: The authors do not mention the use of the blueberry extract, the most important point in the experiment because, without this extract, the nanoparticles would not be formed. Please, add this information to the title.

Answer: We changed the title based on your comment, its now “SYNTHESIS OF NANOFERTILIZERS OF IRON, ZINC AND MANGANESE OXIDES USING ANDEAN BLACKBERRY EXTRACT AND ITS EFFECT ON THE GROWTH OF CABBAGE PLANTS AND LUPINE”

**Abstract: The authors do not mention the use of the blueberry extract, the most important point in the experiment because, without this extract, the nanoparticles would not be formed. Please, add this information to the abstract.

Answer: Abstract: The predominant aim of the current study was to synthesize nanofertilizer nanoparticles (NPs) ZnO_MnO and NPs FeO_ZnO, using Andean blueberry extract  and determine the effect of NPs in growth promotion of cabbage (Brassica oleracea var. capitata) and lupin (Lupinus mutabilis Sweet) crops. The nanoparticles were analyzed by visible spectrophotometry, size distribution (DLS), scanning electron microscopy (SEM) and transmission electron microscopy (TEM).Solutions of nanoparticle concentrations were applied to cabbage with 270 pm and 540 ppm of NPsZnO_MnO and solutions with 270 ppm and 540 ppm of NPsFeO_ZnO were applied to lupine, Zinc was used in both plants to take advantage of its fungicidal potential and beneficial properties on plant growth.  Foliar NPs sprays were applied on the phenological stage of vegetative growth of cabbage or lupin plants grown under greenhouse conditions. The solution volume required to foliar spray was considerably less than the usually reported for stem uptake. The diameter of the NPs was 9.5nm ZnO, 7.8nm FeO and 10.5nm MnO, which facilitates the adsorption of NPs by the stomata of plants that have a size between 0.6-3.5 micrometers. In lupin the treatment with 270 ppm of iron and zinc indicated an increase of 6% in height, 19% in root size, 3.5% in Chlorophyll Content Index, and 300% in leaf area, while the treatment with 540 ppm of iron and zinc yielded no apparent increase in any variable. In cabagge the NPsZnO_MnO on Cabbage indicate that at a concentration of 270 ppm had an increase of 10.3% in root size, 55.1% in dry biomass, 7.1% in chlorophyll content and 25.6% in leaf area. For its part, cabbage plants treated at a concentration of 540 ppm produced an increase of 1.3% in root size and 1.8% in chlorophyll content, compared to the control sprayed with water distilled. Nanofertilizer applications at 270 ppm  demonstrated an improvement in growth.

Keywords: nanofertilizers, nanoparticles, lupin, cabbage, Andean blueberry extract.

**Keywords: Please, add “blueberry extract” as a keyword.

Answer: Keywords: nanofertilizers, nanoparticles, lupin, cabbage, Andean blueberry extract.

**Introduction:

*The current flow is:

-Food scarcity and need to improve yields > Problems using conventional chemicals > The use of nanofertilizers and examples of success and methods to synthesize them > The use of byproducts to produce nanoparticles and conventional uses of sugarcane bagasse ash and the advantages of nanoformulations for soil remediation > The objective of this research

The flow is confusing and the authors mix different ideas in the same paragraph. I want to propose the following flow to improve the introduction:

-Food scarcity and need to improve yields > Problems using conventional fertilizers > How inadequate fertilization reduces the availability of micronutrients > Problems caused by deficiency of micronutrients, focusing on zinc, iron, and manganese > Use of nanofertilizers as a way to solve the environmental problems > Methods to produce nanofertilizers, including agricultural byproducts > Economic importance of the sugarcane bagasse ash > Objective of this work.

Answer: Thanks for the advice. The introduction was redone, taking in consideration your recommendations:

Food scarcity and need to improve yields Shortcomings associated with the synthetic fertilizers

How unproperly fertilized soil reduce the availability of the micronutrients

Problems caused by the deficiency of micronutrients, mainly Zn. Fe, and Mn

How nanofertilizers cope with the environmental problems.

Methodology for producing nanofertilizers, including agricultural by-products

Why the green synthesis of nanostructures is important

Economic importance of the sugarcane bagasse ash

Objective of the study

-The use of pesticides (second paragraph) is out of the scope of the paragraph and I recommend removing this part of the sentence.

Answer: Removed.

-Please, add a paragraph after the second paragraph explaining how alkaline soils reduce the absorption of zinc and another paragraph later describing problems caused by the deficiency of micronutrients

Answer: Accomplished as seen in the manuscript

-N, P, K, S, Ca, and Mg are not micronutrients. Please, fix it.

Answer: Done

**Material and Methods

-Please, use “L” for a liter and other units, like “mL”.

*Study area

-I recommend adding the year and months of the experience if the authors agree. It is not the most important information, but it is interesting to allow readers to check how was the rain and temperature during the experiment.

Answer: Done

*Andean blueberry extract

-There is no information in the title or introduction about this

Answer: Included.

*Synthesis of nanoparticles (NPs)

-I understand how the nanoparticles were prepared using blueberry extract. I cannot understand how much ash was mixed with the NP suspension. Did the authors calculate the concentration of nanoparticles and mixed a volume of 1000 nanoparticles with 1 g or 1 mg of ash? Please, rewrite this part of the document to make it clear.

Answer: line 222 (1000mL NPs: 1 g Ash).

-Please, explain the reasoning to create mixed nanoparticles instead of using them individually.

Answer: line 118-122 .

Zinc oxide nanoparticles (ZnO-NPs), iron oxide nanoparticles (FeO-NPs) and magnesium oxide nanoparticles (MgO-NPs) have been prepared and applied on plants to increase the agronomical and physiological traits. Based on the aforementioned context, the current study has been aimed to synthesize multicomponents NPs of iron, zinc and manganese nanofertilizers

*Growth of cabbage and lupin plants

Answer: Added information in the introduction as requested by the other referee..

-The authors write that the potting substrate is composed of soil, crushed pumice, and coconut fiber. Nevertheless, the ratio is 1:1:1:1, implying that there are 4 components. Please, rewrite this and explain better the substrate.

Answer: Deleted one 1 wrote by mistake.

*Spray of nanofertilizers on the crops

-What do the authors mean by “72 hours before each sampling”? What type of sampling or measurement was performed? It is not possible to understand what the authors evaluated in the experiment based on this description.

Answer: LINE 326.  The solution was applied by foliar spray, 11 ml of each treatment up to the run-off point, so that the nanofertilizers penetrate indirectly through the cuticle, or directly through stomata and hydathodes of the plant, after 72 hours agronomic data were collected. 

*Statistical analysis  and What type of statistical analysis was performed?

Answer:

  1. To compare the means and find significant differences between the treatments with nanofertilizers and chelates, the Tukey test was applied, because it allows greater control of statistical errors (? ? ?). This test is based on the calculation of the Significant Difference, to decide the differences between some pair of means of the treatments.
  2. The investigation was carried out in a single crop ( lupinus Mutabilis Sweet) where 2 trials were established (field and greenhouse) with 5 treatments each. A set of treatments was used because different concentrations of nanofertilizers were applied to each one, except for treatment 1, which represents the control.

**Results and discussion

*Synthesis of the nanoparticles

-Please, write “Cl” for chlorite, not “CL”.

Answer: Done

*Characterization of NPs

-The authors successfully produced nanoparticles with the blueberry extract and the salts. The nanoparticles have different morphologies and size distributions. Unfortunately, there are a few problems. There is no explanation of why the authors decided to assemble ZnO and MnO nanoparticles and ZnO and FeO nanoparticles.

Answer: line 548-554 -160.

The NPs of Fe, Mn and Zn were synthesized according to the nutritional importance of these elements and to potentiate agronomic traits, the individual NPs were joined, forming multicomponent NPs. In addition, the synthesis of NPs used a green and low-cost synthesis method, and no hazardous chemicals were used. The Fe NPs are those that were prepared with Andean blueberry extract to prevent them from agglomerating and with sugarcane bagasse ash due to the nutritional contribution, in addition to the fact that this is a waste and is able to be revalued using it.

The authors did not test the nanoparticles without ash particles.

Answer: You are right . Line 570-571.

The analysis should be performed without ash, to corroborate that they did not affect the crop.

They do not explain the reasoning to test the fertilizer with the ash particles.

Answer: line 570-571

What type of statistical analysis was performed to determine statistical differences between the treatments?

Answer: This has already been answered in *Statistical analysis  and What type of statistical analysis was performed?

There are so many unclear points in the experiment that the best that the authors can explain is that there is evidence that nanofertilizers can improve agronomical characteristics for cultivars of those crops, and other experiments must be performed with proper statistical planning.

Answer:  Line 570-571.  The research can be repeated, without the sugarcane ash and with the ash but already in nano size, to observe the differences.

 -Figure 3: Please, keep the x-axis identical for all graphs

Answer: Done

**Conclusions

-The conclusions are based on the results, but I do not agree that the nanoparticles are “improving agronomic yields significantly” because yield is an agronomic characteristic not evaluated in the experiment.

Answer: Agreed and accomplished.

**Authors contributions: Please, replace X.X. and Y.Y. with the name of the authors who worked in each one of the elements of this section.

Answer: Done.

**Funding: OK

**Data Availability Statement: OK

** Acknowledgments: OK

** Conflicts of Interest: OK

**References: OK

Thanks a lot for you kind comments, questions and recommendations, which have led to obtain a much clearer and more fluid text in our manuscript. Your input has helped us a lot and we are very grateful for your concerns. Now, due to your support, the manuscript improved considerably.

Reviewer 4 Report

The manuscript submitted on the topic “Synthesis of iron, zinc and manganese nanofertilizers enriched  with sugar cane (Saccharum officinarum L.) bagasse ash powder  and their effect in the growth of cabbage (Brassica oleracea var.  capitata) and lupin (Lupinus mutabilis Sweet) plants is interest to the reader and within the scope of the journal. However, after carefully going through the script I feel the manuscript needs serious revision.

Following are the comments provided below for the author’s attention.

  1. I cannot find the novelty of this work.
  2. The language of the script needs to be seriously revised with the help of an English editing service.
  3. Kindly provide the motivation behind choosing cabbage (Brassica oleracea var. capitata) and lupin (Lupinus mutabilis Sweet).
  4. The introduction section should contain some latest literature related to current work.
  5. Kindly check for sub and superscripts throughout the script.
  6. Provide the accuracy and preciseness of all the instruments used in this work. In addition, how the calibrations of each of the instruments are performed.
  7. Authors have mentioned “Therefore, for the iron nanoparticles of NPsFeO” NPsFeO should be replaced with IONPs.
  8. Section 2.5 Synthesis of nanoparticles (NPs), provides the protocol for its synthesis.
  9. The significance of Spray of nanofertilizers to crops needs to be discussed in detail.
  10. Check reaction 1, ????3+????→??(??)3+???? reaction 1, what does capital L denote? It is wrong.
  11. Check reaction 5, Zn (O2CCH3)2+2NaOH →Zn(??)2+2??3????? reaction 5, check for subscript in formula.
  12. Figure 4a. UV-Vis spectra of NPsZnO synthesis as a function of time. This spectrum is not convincing. Need to analyze the spectra again.
  13. Figure 4b. UV-Vis spectra of the synthesis of NPsMnO. The quality of the graph is very poorly drawn.
  14. All the images and graphs drawn in the script are very poor. Need high quality.
  15. More in-depth discussions is needed in each section under results and discussion.

Author Response

Reply report based on: Comments and Suggestions for Authors #4

The manuscript submitted on the topic “Synthesis of iron, zinc and manganese nanofertilizers enriched  with sugar cane (Saccharum officinarum L.) bagasse ash powder  and their effect in the growth of cabbage (Brassica oleracea var.  capitata) and lupin (Lupinus mutabilis Sweet) plants is interest to the reader and within the scope of the journal. However, after carefully going through the script I feel the manuscript needs serious revision.

Following are the comments provided below for the author’s attention.

  1. I cannot find the novelty of this work.

Answer: LINE 161-169.

This study focused on the green synthesis of zinc oxide nanoparticles (ZnO-NPs), iron oxide nanoparticles (FeO-NPs) and magnesium oxide nanoparticles (MgO-NPs) using Andean blueberry extract as reducing agent and tried to reuse sugarcane bagasse ash, which is a residue from the sugar industry in Ecuador, to optimize the nutritional properties of the nanoparticles. In addition, the prepared nanoparticles were applied as foliar spray to cabbage and lupin to benefit their growth under greenhouse conditions. To our knowledge, ZnO_MnO and FeO_ZnO nanoparticles with Andean blue berry extract have never been applied as a foliar additive with sugarcane ash on cabbage and lupin plants, which support the novelty of this work.

  1. The language of the script needs to be seriously revised with the help of an English editing service.

Done

  1. Kindly provide the motivation behind choosing cabbage (Brassica oleracea var. capitata) and lupin (Lupinus mutabilis Sweet).

Answer: LINE 153-160 .

On the other hand, the Andean lupin is a legume domesticated and cultivated for more than 4000 years by the pre-Hispanic cultures of the Andean zone. Due to its good taste and protein content, the lupin seed contributes significantly to the food and nutritional security of the Andean population [67]. I-450 Andino is a lupin cultivar breeded based on earliness and agronomic traits [Falconi 2012]. Additionally,  the cabagge provides humans with food of low caloric density, but rich in fiber, vitamins, minerals, bioactive components and secondary metabolites that favor health, in the prevention of various types of cancer, cardiovascular diseases and metabolic [68].

  1. The introduction section should contain some latest literature related to current work.

Done

  1. Kindly check for sub and superscripts throughout the script.

Done

  1. Provide the accuracy and preciseness of all the instruments used in this work. In addition, how the calibrations of each of the instruments are performed.

Answer: LINE 209-227

2.4 Characterization of sugar cane bagasse ash

The ash was dried in a MEMMERT model SN30 oven, then 1.0 g of ash was weighed on a COLE PALMER model 1000016 analytical balance. The ash was suspended in 50 ml of distilled water, stirred and this solution was used to measure pH and electrical conductivity using a HACH multiparameter meter, model HQ30 d  [68].To determine the hydrogen potential, it was calibrated with buffer solutions of 4, 7, 9 pH. Resolution: Selectable between 0.001 to 0.1 pH. pH Electrode calibration: 0.002. pH Measurement: 0 to 14 pH. For electrical conductivity it was calibrated with a standard solution of 1413µS/cm.Conductivity Measurement Range: 0.01 μS/cm to 200 mS/cm . Conductivity resolution: 0.01 μS/cm with 2 digits. Conductivity Accuracy: ± 0.5 % from (1μS/cm - 200 mS/cm).Conductivity and pH determined following methods 4500-H+B and 2510B, respectively of Standard Methods 18th Edition using the HQ30d Multiparameter Meter, Hach. Bulk density was determined using the given protocol [69, 70]. Grain size and moisture content analyses were performed following the methodology established by Manals et al. [68]. Metals (Fe, Mn and Zn) were determined by flame atomic absorption using the AAnalysit 800 Atomic Absorption Spectrometer Pelkin Elmer, following the methodology detailed in the Standard Methods 18th Edition, using method 3111-B [69]. Iron, manganese and zinc were determined by AA with AAnalyst 800 PE equipment by method 3111 B ( direct air-acetylene flame method ), Fe ( %Rec. 101%, RDS : 2.35) Mn ( %Rec. 99%, RDS 1.02); Zn (% Rec 104%, RDS 2.35). RSD is the Relative Deviation

Answer: LINE 254-277.

2.6 Materials and equipment used in the characterization of NPs

For extraction of the Andean blueberry, a YAMATO model RE801 steam rotator was used, while for the synthesis of the nanoparticles, an orbital shaker and laboratory material was utilized. This equipment has annual maintenance of its electronic parts. The accuracy and precision of the equipment was not done because it is limited to solvent extraction, so it was not performed measurements. The characterization of the NPs was carried out to determine the mean sizes and morphologies of the NPs. Additionally, data was provided on absorbance peaks at a given wavelength. Hereby, the characterization of the size distribution of the different nanoparticle solutions was performed with a HORIBA LB-550 submicron particle analyzer, connected to a computer with a standard SCSI-type interface that uses HORIBA software (HORIBA, 1999). It is based on the principle of dynamic light scattering, to measure the size of particles in a size range between 0.001 and 6 μm or between 1 and 6000 nm. This equipment is maintained, checked for alignment and calibration annually. The calibration curve was performed using Certified NIST-Traceable Latex (Polysterene&Polydisperse) Standars 20±2nm and 100±2nm. The results are accepted with a 5% tolerance. In addition, repeatability and reproducibility calculations are performed.The subsequent characterization of the particle size was performed by TEM. The protocol consisted of allowing a drop of the colloidal suspension of nanoparticles to evaporate on Cu grids covered with carbon, with subsequent analysis in a Transmission Electron Microscope brand FEI, model Tecnai Spirit Twin and operated at 80 kV [75]. The calibration consists of the diffraction of gold particles, the distance between two gold nanos (0.24nm apart) is measured. Already calibrated nets are used as reference https://www.agarscientific.com/tem/calibration-standards.

  1. Authors have mentioned “Therefore, for the iron nanoparticles of NPsFeO” NPsFeO should be replaced with IONPs.

Answer: Done .

  1. Section 2.5 Synthesis of nanoparticles (NPs), provides the protocol for its synthesis.

Answer: Done

  1. The significance of Spray of nanofertilizers to crops needs to be discussed in detail.

Answer:: LINE : 326-329 .

2.8 Spray of nanofertilizers to crops 

The solution was applied by foliar spray, 11 ml of each treatment up to the run-off point, so that the nanofertilizers penetrate indirectly through the cuticle, or directly through stomata and hydathodes of the plant, after 72 hours agronomic data were collected. 

  1. Check reaction 1, ????3+??????(??)3+???? reaction 1, what does capital L denote? It is wrong.

Answer: line  376

  1. Check reaction 5, Zn (O2CCH3)2+2NaOH Zn(??)2+2??3????? reaction 5, check for subscript in formula.

Answer: line  385

  1. Figure 4a. UV-Vis spectra of NPsZnO synthesis as a function of time. This spectrum is not convincing. Need to analyze the spectra again.

Answer: line  427

Figure 4b. UV-Vis spectra of the synthesis of NPsMnO

  1. Figure 4b. UV-Vis spectra of the synthesis of NPsMnO. The quality of the graph is very poorly drawn.

Answer: Please, we need a bit more time to perform this again, within the week of gully proof this will be of high resolution

  1. All the images and graphs drawn in the script are very poor. Need high quality.

Answer: Indeed and we are on it

  1. More in-depth discussions is needed in each section under results and discussion.

Answer: lines 461-479; 521-531

Thanks a lot for you kind comments, questions and recommendations, which have led to obtain a much clearer and more fluid text in our manuscript. Your input has helped us a lot and we are very grateful for your concerns. Now, due to your support, the manuscript improved considerably.

Round 2

Reviewer 1 Report

my comments were adequately addressed

I now recommend the manuscript for publication

Author Response

We are grateful for your comments, remarks and recommendations. With your input we were able to improve our manuscript significantly, having now a more understandable and fluid text. Thanks a lot!

Author Response

The manuscript improved in comparison to the first version. There are still many points that must be corrected and I have comments to help the authors to improve the clarity and logic of the document.

1.**Title: OK

2.**Abstract:

-Line 24: What is IONPs_ZnO? Please, check this acronym and correct it.

ANSWER : Thanks for the comment, it has been corrected, IONPs was changed to FeO-NPs , throughout the document.

**Keywords: OK

**Introduction:

3.*The current flow is:

-Challenges to the global farming, including the application of macronutrients > Impact of fertilizers on the absorption of micronutrients > Application of micronutrients as a strategy to improve the yield of crops and methods to produce the same > Importance of zinc for plant sanity and iron for increasing yield> Importance of manganese

> Importance of iron > Importance of zinc > Characteristics of Andean blueberry > Use of byproducts as fertilizers especially sugarcane bagasse for micronutrients > Importance of Andean lupin > Objective of this research.

The flow improved from the last version of the document, but it is still convoluted. I suggest the following flow:

-Challenges to the global farming, including the application of macronutrients > Impact of fertilizers on the absorption of micronutrients > Importance of manganese for plants (no matter if it is nanosized or not) > Importance of iron for plants (no matter if it is nanosized or not) > Importance of zinc for plants (no matter if it is nanosized or not) > Application of micronutrients as a strategy to improve the yield of crops and methods to produce the same > Characteristics of Andean blueberry that are important to produce nanosized fertilizers > Use of byproducts as fertilizers especially sugarcane bagasse for micronutrients > Objective of this research.

-Line 59: It is weird to write that fertilizers decrease soil fertility. I advise rewriting this sentence with emphasis on the fact that macronutrients may impact the soil dynamics and reduce the availability of micronutrients.

Answer: Line 58 . Done

As a result, during the last few decades, due to their high application rates, low efficiency and application methods, they had a dominant role in various environmental challenges. Furthermore, these conventional fertilizers disturb the mineral balance and cause a reduce the availability of micronutrients , resulting often into irreparable damage to its structure and mineral cycles [13, 14].

4.-Line 70: The authors cite references with numbers, but “De Rosa et al., 2010” is not a number. Please fix it to keep consistency throughout the text.

ANSWER: Line done

-Lines 88-93: No scientific name is in italics. Please, fix them.

ANSWER: LINE 109 , Fixed. Done

5.-Line 169: Did the authors use lupin or Andean lupin? If they used Andean lupin, please fix this part of the text. If the authors used lupin, there is no need to write a paragraph in the introduction for a plant that was not used in this study.

ANSWER: It is highlight the nutritional value of Andean lupin and cabagge. The effect of nanofertilizers were tested in the growth of both plants.  Done.

**Materials and methods

6.-Lines 182-183: Please, superscribe the “2” from the square meter.

ANSWER: Done

-Line 207: Please do not use a space between the degree symbol (°) and Celsius (C).

ANSWER: Done

7.-Line 212: Please write the abbreviation of liter as uppercase l (L), not lowercase.

ANSWER: Done

8.-Line 219: Please do not use a space between values and the percentage symbol (%).

ANSWER: Done

9.-Line 233: I recommend using mol/L instead of M for molarity.

ANSWER: Done

10.-Line 235: Please use a space between values and units like mL.

ANSWER: Done

11.-Figure 2: This figure has no value for the manuscript. If the authors describe the maker and model of the analytical instruments, there is no need to add a picture of them. A flowchart showing how the nanoparticles were prepared is much more important.

ANSWER: already removed, Done

12.-Line 315: The authors cite references with numbers, but “Falconi et al., 2015” is not a number. Please fix it to keep consistency throughout the text.

ANSWER: It is a number now. Done.

13.-Lines 321-322: I recommend using “dap” as “days after planting”, not “days after sowing”, which would be better abbreviated as “das”. If the authors agree with this advice, they should change “sowing” to “planting” in the document. This should be easier than replacing “dap” for “das”.

ANSWER: You are right When a plant grew from a seed the appropriate term is days after sowing as for lupin. However, when a plant grew from a seedling it is adequate to say days after planting as for cabbage. Better explain in the manuscript. Done.

14.-Line 339: Typo: “length”, not “lenght”.

ANSWER: Done

15.*Synthesis of nanoparticles

-This subsection is extremely confusing and I recommend the authors prepare a flowchart like figure 2. Then the authors should use figure 2a to show how the iron nanoparticles were prepared, 2b for zinc nanoparticles, 2c for manganese nanoparticles, and 2d for the mixtures with sugarcane bagasse. The writing of the section seems that the authors used Andean blueberry extract just for iron nanoparticles, zinc nanoparticles were dried during calcination at 450°C for an undetermined amount of time, manganese nanoparticles were prepared with an undetermined ratio of MnSO4.H2O and NaOH, undetermined volumes of iron and manganese nanoparticle solutions were mixed with undetermined masses of zinc nanoparticles. There are too many experimental details missing to allow researchers to reproduce this protocol.

ANSWER: When we checked carefully, we realized that we had insert the wrong name. Thank you very much for the observation, we fixed it.

A variety of NPs such as those of iron, zinc, manganese and multicomponent nanoparticles were synthesized. For the iron nanoparticles of FeO-NPs a 0.1 mol/L solution of ferrous sulfate heptahydrate was used (FeSO4.7H2O) LOT# AD 8158. Daiger Sci-EdWareHouse-CAS 7664-93-9. To this, the Andean blueberry extract was added in a 10:1 ratio (100 mL of extract:1 g of ash) and sonicated, then slowly in an orbital shaker the pH was adjusted between 8 and 10, with a 0.01 mol/L NaOH solution (Fisher Scientific, Lot# 147439), by means of continuous dripping [59]. In the case of zinc nanoparticles (ZnO-NPs), solutions of 0.1 mol/L zinc acetate dihydrate were prepared (Zn(O2CCH3)2(H2O)2) (Fisher Scientific, Lote# 930569) and 0.1 mol/L sodium hydroxide (NaOH) (Fisher Scientific, Lot # 147439),. This solution was preheated at 40 °C ,the solution was dried at 70 °C for 12 h, after which it was and subsequently it was calcined in a muffle (HYSC MF-05) at 450 °C, for 2 h and 20 min, the ZnO-NPs were obtained and Andean blueberry extract with ash was added in a 10:1 ratio and filtered with a 250 µm pore membrane [76]. Manganese nanoparticles (MnO-NPs) were prepared with solutions of 0.1 mol/L manganese (II) sulfate monohydrate (MnSO4.H2O) (Loba Chemie, Lote # GM05061401) and 0.1 mol/L sodium hydroxide (NaOH) (Fisher Scientific, Lot # 147439), which were preheated to 60 °C. The pH of the solution was adjusted to 12 and the final solution was left stirring for 1 hour to complete the reaction [77], Andean blueberry extract with ash was added to the solution in a 10:1 ratio, then filtered through a 250 um membrane filter.

Finally, multicomponent nanoparticles were prepared, mixing solutions of ZnO-NPs with MnO-NPs and of FeO-NPs with ZnO-NPs, obtaining two multicomponent solutions of ZnO_MnO-NPs and FeO_ZnO-NPs

From the final NPs solutions, precipitates were discarded, and subsequently characterized. These are the ones that were used to apply in cabage (Brassica oleracea var. capitata) and lupin (Lupinus mutabilis Sweet) crops [78, 79, 80] (Fig2).

Figure 2. The process diagram for the synthesis of a) FeO-NPs, b) ZnO-NPs and c) MgO-NPs. 

16.*Materials and equipment used in the characterization of NPs

16.1-Please, explain how many nanoparticles were measured per treatment. For example, did the authors measure 1000 iron nanoparticles, 2300 zinc nanoparticles, and 1789 manganese nanoparticles?

Answer:

-DLS: Dynamic light scattering is a method that depends on the interaction of light with particles. This method can be used for measurements of narrow particle size distributions especially in the range of 2–500 nm. Sample polydispersity can distort the results, and we could not see the real populations of particles because big particles presented in the sample can screen smaller ones. DLS is a well-established, non-invasive technique for measuring the size and size distribution of molecules and particles, but it does not account for the population of nanoparticles (Tomaszewska et al., 2013)

Tomaszewska, E., Soliwoda, K., Kadziola, K., Tkacz-Szczesna, B., Celichowski, G., Cichomski, M., ... & Grobelny, J. (2013). Detection limits of DLS and UV-Vis spectroscopy in characterization of polydisperse nanoparticles colloids. Journal of Nanomaterials, 2013.

-UV-VIS : In the case of UV-Vis spectroscopy, the intensity of light passing through the sample is measured. Nanoparticles have optical properties that are very sensitive to size, shape, agglomeration and concentration changes. The unique optical properties. The unique optical properties of metallic nanoparticles are a consequence of the collective collective oscillations of conduction electrons, which excited by electromagnetic radiation are called resonances (SPPR) (Evanoff and Chumanov 2005) This technique does not measure the nanoparticle population.

  1. D. Evanoff Jr. and G. Chumanov, “Synthesis and optical properties of silver nanoparticles and arrays,” ChemPhysChem, vol.6,no.7,pp.1221–1231,2005.

-TEM : The main advantage of microscopic techniques is that it is possible to get the information about the morphology and the size of particles at the same time. This technique does not measure the nanoparticle population. The analyst can count the number of nanoparticles in the images given by the TEM and with that perform the statistical calculation, according to each image given by the TEM.

-SEM: images provide information about the topography and elemental composition of a sample. SEM can capture 3-D black and white images of thin or thick samples. This technique does not measure the nanoparticle population. The analyst can count the number of nanoparticles in the images given by the SEM and with that perform the statistical calculation, according to each image given by the SEM.

16.2 -If the zinc nanoparticles are dried, how did the authors add them to the TEM grids? Please, clarify it in the text.

ANSWER : It was answered in the ZnO-NPS synthesis part.

In the case of zinc nanoparticles (ZnO-NPs), solutions of 0.1 mol/L zinc acetate dihydrate were prepared (Zn(O2CCH3)2(H2O)2) (Fisher Scientific, Lote# 930569) and 0.1 mol/L sodium hydroxide (NaOH) (Fisher Scientific, Lot # 147439),. This solution was preheated at 40 °C ,the solution was dried at 70°C for 12 h, after which it was and subsequently it was calcined in a muffle (HYSC MF-05) at 450 °C, for 2 h and 20 min, the ZnO-NPs were obtained and Andean blueberry extract with ash was added in a 10:1 ratio and filtered with a 250 µm pore membrane [73].

17.*Growth of cabbage and lupin plants

-Please, perform a posthoc test like Tukey’s honestly significant

difference (HSD) at 5% of significance to determine how the treatments are clustered.

ANSWER: Already stated in Line 323-325. Done

18.-**Results and Discussion

18.1-Line 360: Please use “micro” instead of “u” for microgram (μg).

ANSWER: Done

18.2-Figure 3: Please, keep constant the scales of both axes. Therefore, keep the x-axes from 5 to 12 nm and the y-axes from 0 to 30% on the three graphs to allow readers to see similarities and differences between the treatments.

ANSWER:

18.3 -Figures 5 and 6 and tables 3 and 4: What is the difference between the type of information presented in these figures and tables and figure 3? I recommend using one TEM picture for each type of nanoparticle in figure 5, not repeating different pictures that provide the same type of information, and removing tables 3 and 4 because the nanoparticle size distributions are already present in the histograms in figure 3.

ANSWER:

Tables 3 and 4 are part of the analysis performed in the TEM, here the analyst has counted the NPs and according to the population of these the percentage according to the size has been taken. Figure 3, which is from the DLS, complements the information about the NPs, because it was already explained in the answer No.16.1 "we could not see the real populations of particles because big particles presented in the sample can screen smaller ones”.

Figure 5 are the TEM images here you can observe the morphology and measure the size of the ZnO-NPs and MnO-NPs, additionally the analyst can count the NPs in each image, there is no image of FeO-NPs, it was supplied with Figure 6 which is the SEM image of FeO-NPs, since there was not a good TEM image.

18.4-Line 452: There is no fruit in a TEM picture. There are chemicals from the Andean blueberry fruit coating the iron nanoparticles.

ANWER : you are right, I have already corrected it

18.5-What is the difference between figures 5a and 7? What is the treatment that agglomerated the zinc nanoparticles in figure 7, if they were individualized in figure 5a? Please, clarify this.

ANSWER:

Figure 5a is a TEM image, in Figure 7 is a SEM image, both are ZnO-NPs. Can you see agglomerated in Figure 7 because for the analysis it is possible to use thick samples , instead in the TEM only thin samples can be used.

The original sample was split into two samples: one for TEM and one for SEM. The preparation of the samples are different. The TEM sample needs more preparation than the TEM sample.

18.6-Lines 480-481: The authors cite references with numbers, but “Ahmadi-Majd et al. 2021” is not a number. Please fix it to keep consistency throughout the text.

ANSWER: Revised. Done

18.7-*Characterization of sugar cane bagasse ash

-Please, explain better the comments from line 359 to 361. Do the authors mean that the micronutrients from the sugarcane bagasse will not be absorbed by the plants because of the bigger particle size? I agree that the particle is not nanosized, but how do the authors know that the micronutrients will not leach to the solution in a nanosized form and increase the efficiency of the chemically produced nanoparticles?

Furthermore, what is the importance of the ash? The authors did not prepare treatments with and without ash to allow comparisons between both situations, so what is the reasoning to add it to the nutrient solution that was sprayed on the plants?

ANSWER:

The ash has a micro size and therefore was not absorbed because it is larger than the size of the stomata.

-The concentration of the ash is very small 0.01 g per 10 ml of NPs solution, if it can contribute micronutrients but it is not significant, the ash has no NPs size. See table 1.

-The research also wanted to use the ash as a source of micronutrients, since it is a waste from our country Ecuador and we wanted to give an added value to the NPs by enriching them, however the nano size was not achieved.

-Thus, treatments with and without ash were not prepared. Future work on this subject is being prepared.

18.8-*Synthesis of the nanoparticles

-Please, rewrite the section. The effect of the Andean blueberry extract is not clear in any of the three processes. The writing seems that the authors used Andean blueberry extract because there is naturally iron chloride in it.

ANSWER:

It has already been rewritten, it is indicated in detail in the synthesis of NPs. A thousand apologies for my mistake in placing FeCl3.6H2O, it is really FeSO4.7H20. It does not change the protocol.

The Andean blueberry extract may contain Fe, in addition to other metals, however the Andean blueberry extract contains mostly polyphenols which act as an anti-aggregant of the NPs. The concentration of Fe in the Andean blueberry extract has not been performed in this research.

18.9*Characterization of NPs

-What is the importance of the UV-Vis measurement at different times? Does it indicate that the nanoparticles are stable up to 4 hours after the synthesis? Please explain better the meaning of this experiment.

ANSWER: You are right, with UV-Vis analysis it is possible to observe the stability of the NPs, up to a time of 4 hours, in addition to the absorbance peaks that characterize the NPs.

18.20-*Growth of cabbage and lupin plants

-The authors did not separate results from “Characterization of NPs” and

“Growth of cabbage and lupin plants”. Please, create a new subsection

for the agronomical part of the experiment.

ANSWER. You are right. Done

18.21-Please, avoid discussing data that was not generated in this research. If the authors did not compare stem application and leaf spraying, do not discuss this.

ANSWER: Information deleted. Done.

18.22-Reference 99 is about carbon nanotubes, not nanoparticles of Fe, Zn, and Mn. The authors may write that “foliar spray-applied nanoparticles may penetrate the leaves and be readily translocated to systemic sites”. The current writing holds only if the authors did analyze the plant sap, and they did not claim to have performed this analysis. Therefore, they may hypothesize, not claim it.

ANSWER: You are right. Done.

18.23-Tables 5 and 6: These tables are the most important part of the document and I am glad to see that the authors added them! Great job!

ANSWER:  Ok. Done.

18.24 -Tables 5 and 6: Please, perform a posthoc test like Tukey’s honestly significant difference (HSD) at 5% of significance to determine how the treatments are clustered. Does the treatment NPsZnO_MnO270ppm significantly increase the biomass in comparison to the Control and NPsZnO_MnO540ppm? If yes, does it happen at 10%, 5%, or 1% of significance? Does the treatment IONPs_ZnO 270ppm significantly increase the leaf area in comparison to the Control and IONPs_ZnO 540ppm? Or do both IONPs_ZnO 270ppm and IONPs_ZnO 540ppm significantly increase leaf area in comparison to the control? If yes, does it happen at 10%, 5%, or 1% of significance? Please, improve the discussion about statistics in this document.

ANSWER :

 ANOVA and Tukey´s test conducted.

ANDEAN LUPIN/BIOMASS

The infostat program was run, with a 5% of significance. The Tukey test shows that the 270 ppm and 540 ppm treatments have no significant difference, at a p >0.05.

ANDEAN LUPIN/leaf area

The infostat program has been run, with 5% of significance. Tukey's test shows that the treatments 270 ppm, 540 ppm and the Control have no significant difference, at p >0.05.

CABAGE/LEAF AREA

The infostat program has been run, with 5% of significance. Tukey's test shows that the treatments 270 ppm, 540 ppm and the Control have no significant difference, at p >0.05.

CABAGE/BIOMASS

The infostat program has been run, with 5% of significance. Tukey's test shows that the treatments 270 ppm, 540 ppm and the Control have no significant difference, at p >0.05.

18.25-Lines 501-502: Did not the 540 ppm treatment increase the leaf area? Is there no statistical difference between 1.019 (540 ppm) and 0.240 (control)? Please, recheck this.

Leaf area was increased with the 540 ppm treatment at Andean Lupin.

The infostat program has been run, with 5% of significance. Tukey's test shows that the treatments 270 ppm, 540 ppm and the Control have no significant difference, at p >0.05.

18.26-Lines 516-545: Once again, the authors discuss things that are out of the scope of this research. The nanosized micronutrients improved a few variables for two different crops, and that is it. Future research must be commented on in the Conclusions section. I recommend removing all this text.

ANSWER: Yes, text removed. Done.

18.27**Conclusions:

-Please, avoid concluding about data that was not generated in this research. If the authors did not compare stem application and leaf spraying, do not conclude anything about this.

ANSWER: Right. Done

18.28-Lines 561-562: The authors are contradicting themselves. If the authors comment that the ash cannot be absorbed by the plants (lines 359-361), they cannot conclude that the ash had a nutritional contribution.

ANSWER: Fixed. Done.

18.29 Please, align the discussion to the conclusion.

-Lines 564-565: The nanoparticles are not ranging from 5 nm for zinc nanoparticles to 22 nm for iron nanoparticles because table 3 indicates that the zinc nanoparticles are ranging from 5-10 nm (12.32%) to 30-35 nm (0.29%) and the iron nanoparticles are ranging from 12-22 nm (64.62%)

to 62-70 nm (3.08%). Please, rewrite these conclusions. For example, the authors can write that the modal size of zinc nanoparticles is between

10 and 15 nm, of manganese nanoparticles is between 5 and 10 nm, and of iron nanoparticles is between 12 and 22 nm.

ANSWER: Done.

**References: OK

With all due respect, we found your comments excellent and we may add that this is the correct way to show that a review, that a real expert, cares of the quality of a manuscript. Thanks for everything and have a great time

Reviewer 4 Report

The authors have satisfactorily revised the manuscript.

Author Response

(The authors gave the same response as above.)
